# Programmable self-regulated molecular buffers for precise sustained drug delivery

Arnaud Desrosiers[1,2], Rabeb Mouna Derbali[3], Sami Hassine[2], Jérémie Berdugo[4], Valérie Long[3,5], Dominic Lauzon[1], Vincent De Guire[6], Céline Fiset [6][3,5], Luc DesGroseillers [6][2], Jeanne Leblond Chain[7] & Alexis Vallée-Bélisle [6][1,2] ✉

Unlike artificial nanosystems, biological systems are ideally engineered to respond to their environment. As such, natural molecular buffers ensure precise and quantitative delivery of specific molecules through self-regulated mechanisms based on *Le Chatelier's* principle. Here, we apply this principle to design self-regulated nucleic acid molecular buffers for the chemotherapeutic drug doxorubicin and the antimalarial agent quinine. We show that these aptamer-based buffers can be programmed to maintain any specific desired concentration of free drug both in vitro and in vivo and enable the optimization of the chemical stability, partition coefficient, pharmacokinetics and biodistribution of the drug. These programmable buffers can be built from any polymer and should improve patient therapeutic outcome by enhancing drug activity and minimizing adverse effects and dosage frequency.

One of the key factors to achieve successful disease treatment is to provide and maintain a therapeutic drug dosage throughout treatment. Sub- or overtherapeutic exposure reduces treatment efficiency, the former leading to drug resistance and the latter increasing side effects[1–3]. Maintaining an optimal therapeutic concentration at the target site, however, remains a major challenge in modern medicine for several reasons. First, most drugs undergo rapid degradation/clearance[4], forcing patients to take multiple doses at regular intervals during the course of their treatment. This repeated dosage regimen typically leads to poor compliance[5] and is responsible for 33–69% of medication-related hospital admissions in the USA[6]. To simplify and improve therapeutic compliance, various kinetically programmed drug delivery systems (DDS) have been developed over the years[7–10]. Typically, an oral or local sustained release is achieved using erodible or swelling polymer matrices that delay the diffusion of a large drug payload, compensating for the drug degradation/clearance[7]. Injectable DDS, such as lipid or polymer-based nanomedicines, encapsulate drugs with unfavorable biopharmaceutical properties (low solubility,

low permeability) and improve their bioavailability, biodistribution, and usually prolong their blood circulation time[7]. Unfortunately, these DDS do not take into account the individual pharmacokinetic specificities and result in significant interindividual variability in drug plasmatic concentrations[11,12]. Impressive progress has been made in the local administration of on-demand drug delivery systems, for instance, intelligent wearable medical devices, but their development is still complex and costly[9,13]. Furthermore, these DDS cannot prevent drug overdosing, which killed more than 70,000 people in the USA in 2019[14]. As we enter the era of smart DDS[15], novel systems solving these challenges will have a major impact on the success of medical treatments.

Nature has evolved various mechanisms to achieve optimal self-regulated dosing of molecules regardless of an individual-specific pharmacokinetic profile. Protein transporters, for example, act as molecular buffer agents to maintain a precise concentration of free active molecules using a mechanism analogous to pH buffers. Thyroxine-binding globulin, TBG, for example, circulates in the blood

[1]Laboratoire de Biosenseurs et Nanomachines, Département de Chimie, Université de Montréal, Montréal, QC H3C 3J7, Canada. [2]Département de Biochimie et Médecine Moléculaire, Université de Montréal, Montréal, QC H3T 1J4, Canada. [3]Faculté de Pharmacie, Université de Montréal, PO Box 6128 Downtown Station Montréal, QC H3C 3J7, Canada. [4]Département de Pathologie, Université de Montréal, Montréal, QC H3T 1J4, Canada. [5]Centre de Recherche, Institut de Cardiologie de Montréal, Montréal, QC H1Y 3G4, Canada. [6]Clinical Biochemistry Department, Maisonneuve-Rosemont Hospital, Optilab-CHUM Laboratory Network, Montreal, QC, Canada. [7]Univ. Bordeaux, CNRS, INSERM, ARNA, UMR 5320, U1212, F-33000 Bordeaux, France. ✉e-mail: a.vallee-belisle@umontreal.ca

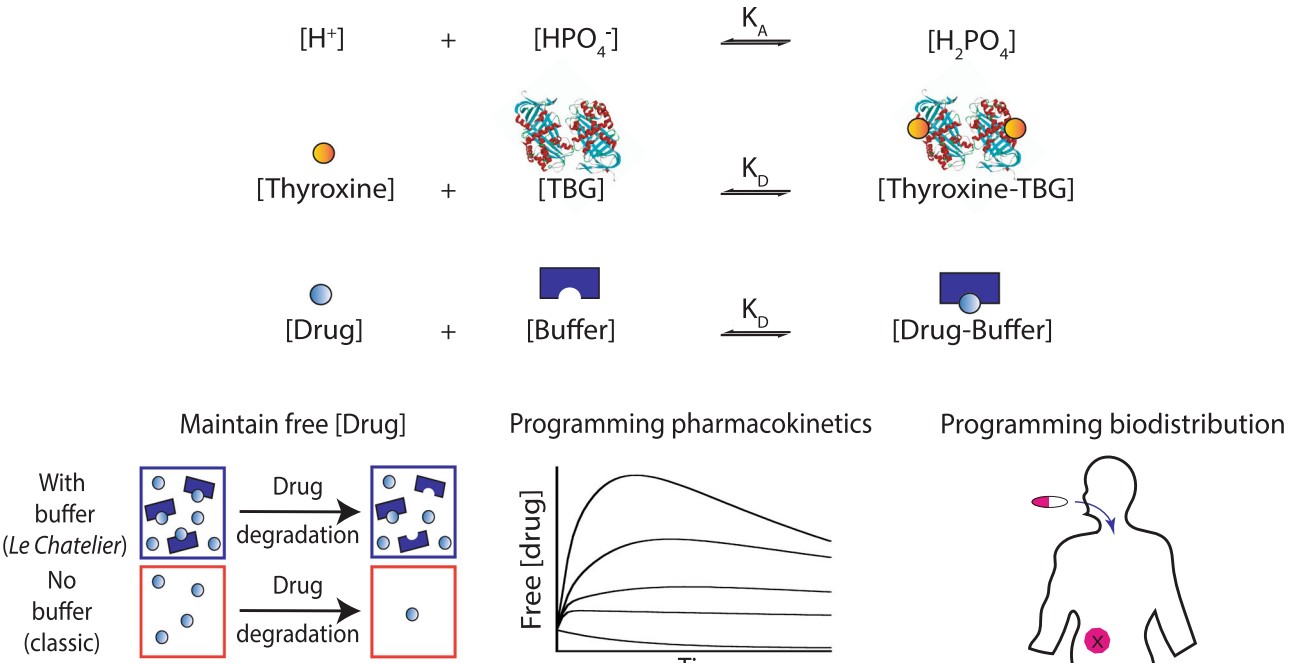

**Fig. 1 | Drug delivery systems (DDS) based on natural molecular buffers.** Top: Nature employs molecular buffers like $HPO_4^-$ and TBG to maintain a constant concentration of active biomolecules using *Le Chatelier's* principle for $H^+$ and thyroxine, respectively. Bottom left: Similarly, we can engineer self-regulated molecular buffers to sequester a large reservoir of inactive drugs and to maintain the free active drug concentration at a precise concentration despite drug degradation over time by releasing the bound drugs. Bottom middle and right: We hypothesize that drug pharmacokinetics and biodistribution can be modulated by tuning the molecular properties of the buffer.

at a high concentration of 270 nM and transports a large reservoir of inactive thyroxine molecules of 100 nM, while maintaining the free active thyroxine concentration at 22.5 pM[16]. Such molecular buffer follows *Le Chatelier's* principle, in which the equilibrium between the complexed and free thyroxine shifts in response to changes in thyroxine levels in the blood to maintain its free concentration near the thyroxine-TBG dissociation constant $K_D$ ~100 pM[16]. Currently, sustained release systems do not follow this principle, since their release is not impacted by the elimination of the drug. Few self-regulated devices or platforms have been developed, but they have not yet reached the clinics[15]. The active or target concentration required for the desired therapeutic effect is typically obtained through pharmacodynamics studies during preclinical and clinical studies. Then the therapeutic dosage is selected and adjusted in order to reach optimal therapeutic outcome[17]. However, the selection of an optimal therapeutic dosage for a new drug entity remains a major challenge since 16% of drugs fail the FDA review cycle because of uncertainties in their dosage selection[18]. Strategies overcoming this challenge by delivering the drug directly at the target concentration would greatly improve the chances of success of new drug candidates. In addition, the ability to maintain the therapeutic drug concentration in patients displaying different pharmacokinetics would drastically reduce the inter-individual variability and enlarge the target patient population. Furthermore, it would likely reduce the frequency of administration, and improve patient compliance and treatment efficacy.

In this work, we developed bio-inspired programmable drug buffers with self-regulating properties (Fig. 1). We show that these programmable buffers maintain a constant concentration of free drugs and enable their pharmacokinetics and biodistribution to be tuned (Fig. 1).

## Results

### Designing and programming the molecular buffers

As a proof of concept, we have engineered DNA-based drug buffers for the antimalarial drug quinine and the chemotherapeutic drug doxorubicin. We have built these buffers using DNA chemistry since: (1) like antibodies, DNA molecules can be selected to bind various molecular targets with high selectivity and specificity like aptamers[19–21]; (2) unlike antibodies, DNA synthesis is simple, inexpensive[22], and supports the simple addition of various modification groups that can stabilize and drastically increase DNA half-life in blood circulation[23]; (3) the simple base-pairing code of DNA enables easy tuning of its binding affinity for their target[24–27]; (4) nucleic acids are being increasingly used as drugs or drug carriers[28–32] and (5) the fluorescence of quinine and doxorubicin is quenched upon binding to their DNA-binding sequence[33–36], enabling efficient quantification of the free drug concentration (Supplementary Fig. 1).

We first demonstrated the capacity of our molecular buffer to maintain a precise free drug concentration despite large variations in total drug concentration. This buffer capacity was first highlighted by Van Slyke in 1922[37]. The buffer capacity of a molecular buffer can be easily determined using a titration curve, the equivalent of the pH buffers titration curves, which displays a buffer zone near the $K_D$ in which the free drug concentration remains relatively constant upon increasing the total drug concentration (supplementary Fig. 2). For our quinine buffer, we employed a quinine-binding DNA aptamer, Q0, that displays a dissociation constant, $K_D$, of 90 nM (supplementary Fig. 3A and supplementary Tables 1, 2)[33]. For our doxorubicin buffer, we employed a DNA-binding sequence, D0, that displays a $K_D$ of 130 nM (Supplementary Fig. 3B and supplementary Tables 1, 2)[34]. The drug/DNA buffer stoichiometries were determined to be $0.95 \pm 0.06$ and $2.7 \pm 0.4$ for Q0 and D0, respectively (supplementary Fig. 3C and supplementary Table 2). As expected, we found that the buffer exhibits its optimal buffer capacity β when the free drug concentration matches the dissociation constant value: $\beta^{max}_{quinine} = 135 \pm 21$ nM and $\beta^{max}_{doxo} = 128 \pm 7$ nM (Fig. 2A, B and E, F, blue dotted line and supplementary Fig. 4A, B). For example, in the presence of a 20 μM buffer, we can maintain the concentration of free quinine in the hundred-nanomolar range, even when the total quinine concentration is as high as 15 μM (Fig. 2B–F).

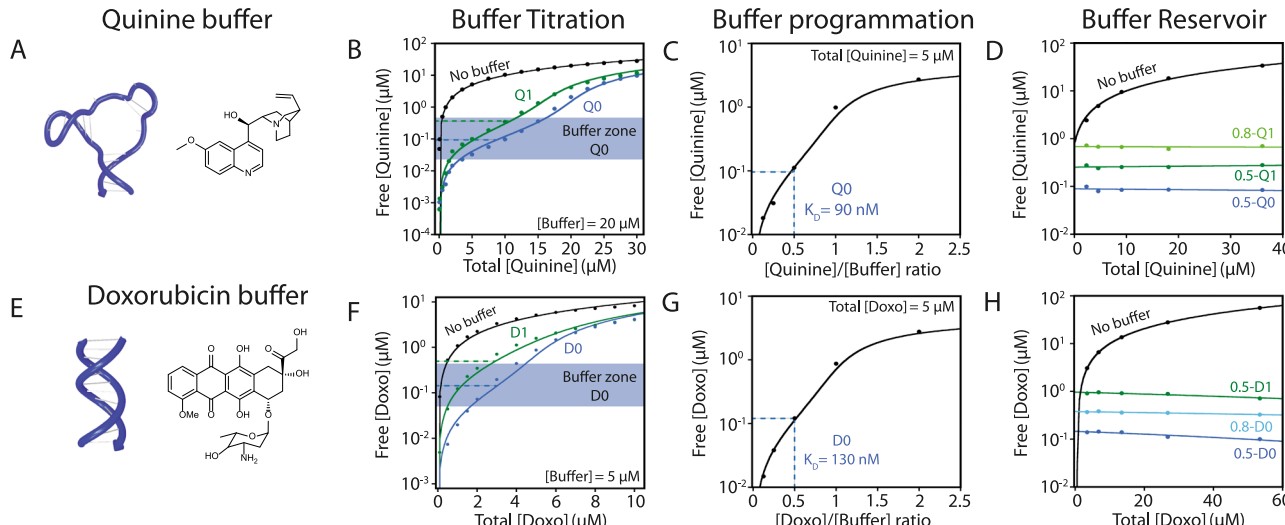

**Fig. 2 | Programming the buffer capacity of molecular buffers.** Quinine (**A**–**D**) and doxorubicin (**E**–**H**) buffers maintain the concentration of free quinine and free doxorubicin near their $K_D$ values even in a large drug concentration range. **B**, **F** Increasing the dissociation constant of the buffer, $K_D$, increases the concentration of free drug proportionally with buffer variants Q1 and D1 for example (black line: no buffer added; green line: D1 and Q1 buffer variant; blue line: D0 and Q0 buffer variant). Data fitted with equation E5—see Supplementary Information. **C**, **G** The free drug concentration can also be precisely programmed by varying the

[Drug]/[Buffer] ratio. Data fitted with equation E5—see Supplementary Information. **D**, **H** The free drug concentration can be maintained over large variations of total drug concentration by maintaining a constant [Drug]/[Buffer] ratio (black line: no buffer added; light green line: 0.8-Q1; green line: 0.5-Q1 and 0.5-D1; blue line: 0.5-Q0 and 0.5-D0; light blue line: 0.8-D0). Data fitted with linear regression. All data were obtained using fluorescence measurements (Supplementary Fig. 1), and the errors were obtained from the fit.

We can also program our buffers to maintain a specific, desired concentration of a free drug. The first approach consists of varying the $K_D$ of the buffer. For the quinine aptamer, we introduced site-specific mutations (see Materials and Methods) that reduce the affinity for quinine (aptamer Q1 and Q2). Since doxorubicin binds duplex DNA through intercalation in GC base pair, mutations could not be used to tune their $K_D$. To circumvent this limitation, we explored and found that specific G-quadruplex sequences (D1 and D2) display $K_D$ that are 4 and 27-fold higher than the original D0 GC duplex DNA[35,36]. For example, buffer variants Q1 and D1 (see supplementary Fig. 3) are engineered to display lower affinities for quinine and doxorubicin with $K_D$ of 198 and 703 nM, respectively, which moves the free drug concentration to higher values while maintaining a similar buffer capacity (Fig. 2B–F, green dotted line). As expected, we find that the optimal buffer capacity of both buffers is reached when the free drug concentration is equal to the $K_D$ (supplementary Fig. 4). However, we can also modify the [Drug]/[Buffer] ratio to further tune the free drug concentration at equilibrium. For example, a [Doxo]/[D0] ratio of 1 maintains the concentration of free doxorubicin to 10x of the value of the $K_D$, while a ratio of 0.1 maintains the concentration of free doxorubicin to 0.1x of the value of the $K_D$ (Fig. 2C–G and equation E5—see Supplementary Information). We then demonstrated the capacity of specific [Drug]/[Buffer] formulations, like 0.5-Q0, 0.5-Q1, or 0.8-Q1, to maintain a specific concentration of free drug even when the concentration of the formulation is varied by over 22-fold (Fig. 2D–H). As demonstrated later, this feature can be exploited to make sure patients remain in the therapeutic window for an extended period of time, regardless of the amount of the formulation administered.

### In vitro proof of concept of molecular buffers
The reservoir capacity of the proposed buffer suggests that significantly higher drug amounts could be administered in a single dose. However, this raises toxicity issues, and we sought to ensure that the buffer completely inactivates the sequestered drug using two cancer cell lines: HeLa and HCT116. We first determined the toxicities of doxorubicin for HeLa cells with IC$_{50}$ = 680 nM (Fig. 3A) and for HCT116 cells with IC$_{50}$ = 528 nM (supplementary Fig. 5A), which were similar to

the values reported in the previous work[27]. We also confirmed that molecular buffers alone were well tolerated by both cell types, which was expected due to the nucleic acid nature of the buffers (supplementary Fig. 6). We then demonstrated the capacity of a [Doxo]/[D0] formulation to provide specific free doxorubicin concentrations by measuring cell viability (Fig. 3B and supplementary Fig. 5B). The 0.5-D0 formulation, programmed to achieve a free doxorubicin concentration near the IC$_{50}$ with its $K_D$ = 968 nM at 37 °C, maintained a cell viability level close to 50% even in presence of 50 μM of total doxorubicin (Fig. 3B). Similarly, decreasing the [Doxo]/[D0] ratio to 0.2, which decreases the free doxorubicin concentration to 300 nM, provided higher viability levels up to 80% for the same wide range of formulation concentrations (Fig. 3B and supplementary Fig. 5B). Using flow cytometry and confocal microscopy, we showed that cells treated with 10 μM of a 0.5 [Doxo]/[D0] formulation, programmed to deliver 1 μM of free doxorubicin, display similar levels of doxorubicin cellular uptake (Fig. 3C and see supplementary Fig. 7 for gating strategy) and nuclear localization (Fig. 3D, E) as cells treated directly with 1 μM of free doxorubicin. Interestingly, free doxorubicin is detected mainly in the nucleus, where it binds to genomic DNA, even though the molecular buffers were only detected in the cytoplasm of HeLa cells (supplementary Fig. 8).

### Programming drug reservoir and its kinetic release
In addition to maintaining free drug concentration, the molecular buffers can also act as drug reservoirs that can significantly reduce the frequency of administration of medical treatment. To illustrate this advantage, we simulated a drug/buffer formulation degradation or elimination through serial ½ dilutions every hour that mimics a drug/buffer concentration decrease by 50% every hour. (Fig. 4A, left panel). Of note, doxorubicin associates and dissociates from its DNA buffer within less than 10 milliseconds (supplementary Fig. 9), allowing this drug-buffer reservoir to always remain in equilibrium. In the absence of buffer, the first dilution decreases the free doxorubicin concentration by half. In the presence of buffer formulation 0.5-D0, the reservoir effect maintains the free doxorubicin concentration within an arbitrary therapeutic window at 75–150 nM (Fig. 4A middle panel, blue

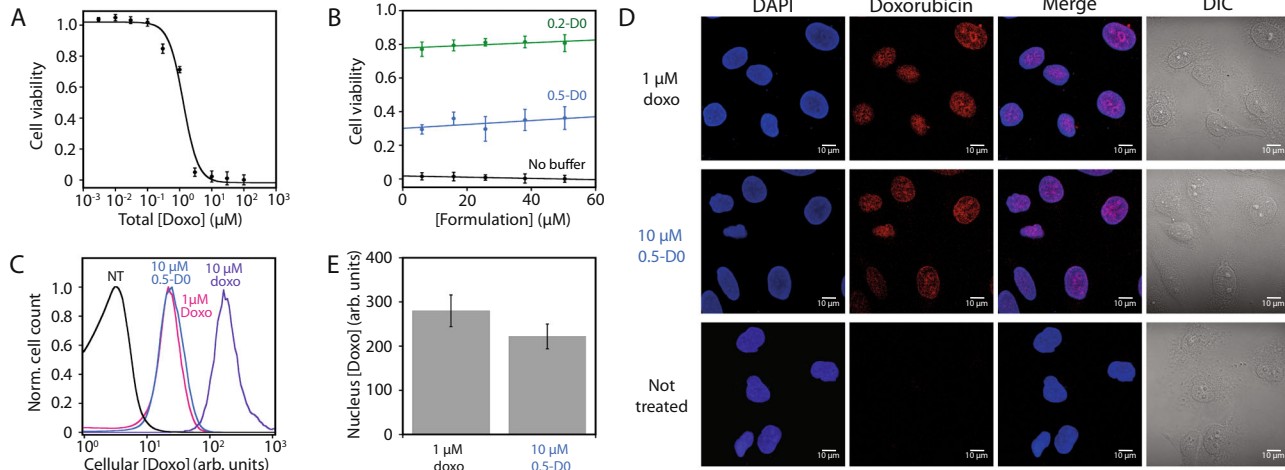

**Fig. 3 | In vitro validation of the doxorubicin programmable molecular buffer (D0). A** Cytotoxicity of doxorubicin on HeLa cells was assayed with resazurin ($n = 8$ biologically independent samples). Data fitted with equation E21 (see Supplementary Information) and are presented as mean values ± SD. **B** The 0.5-D0 (blue line) and 0.2-D0 (green line) formulations, programmed to maintain drug concentration near 1 μM and 300 nM, respectively (see supplementary Table 2 and equation E9), provided a cell viability level matching these free drug concentrations even at a very high doxorubicin concentration of 50 μM ($n = 8$ biologically independent samples). Data fitted with linear regression are presented as mean values ± SD. **C** HeLa cellular uptake levels of 1 μM (pink line) or 10 μM [Doxo] (purple line) and 10 μM 0.5-D0 (blue line) demonstrate that only the free doxorubicin fraction is taken up by the cell (flow cytometry, min 20,000 events/measurement; black line: no treatment). **D** Doxorubicin nuclear localization by confocal microscopy with nuclei stained by DAPI. The 10 μM 0.5-D0 formulation displays similar doxorubicin cellular distribution in the nucleus as 1 μM doxorubicin in the absence of buffer. For each condition, $n = 3$ independent experiments. **E** Quantification of nuclear localization by ImageJ ($n = 100$ cells examined over three independent experiments). Data were presented as mean values ± SD.

rectangle). By increasing the reservoir concentration, we can prolong the therapeutic exposure (time within therapeutic window) of a drug by up to ninefold with 160 μM of the 0.5-D0 buffer reservoir (Fig. 4A right panel).

Molecular buffers can also be programmed to generate personalized pharmacokinetic profiles by controlling their degradation or elimination rate. To demonstrate this strategy, we designed an experimental setup simulating doxorubicin degradation/elimination by the kidneys using a dialysis cassette in which the doxorubicin that has passed through the membrane outside the cassette is considered degraded/eliminated (Fig. 4B, left panel). We also programmed the buffer degradation rate using specific amounts of DNAse I (supplementary Fig. 10). In this setup, we measured the free doxorubicin concentration inside the dialysis cassette by fluorescence in the presence of 10 μM of a 0.5-D0 formulation which corresponds to 10 μM of doxorubicin and 20 μM of D0 DNA buffer. In the absence of buffer, half of the doxorubicin was released from the cassette in the first 4−5 h (supplementary Fig. 11). When employing a 0.5-D0 formulation at a very low concentration of DNAses of 0.098 U/mL, most doxorubicin remained bound to its buffer, which drastically reduced the release rate of doxorubicin from the cassette with an estimated 50% doxo released in ~160 hrs. Increasing the degradation rate of buffer D0 with an increasing concentration of DNAse I led to an increase of the free drug concentration and a decrease in the [Doxo]/[D0] ratio over time (Fig. 4B middle panel and see equation E18). This increase in free doxorubicin concentration led to a faster release from the dialysis bag, resulting in an apparent shorter half-life of doxorubicin (Fig. 4B right panel). These results demonstrate how one can create custom-made pharmacokinetic profiles by simply programming the buffer degradation rate.

### Tuning drug pharmacological properties through molecular buffers optimization

We can also tune the pharmacokinetic properties of a drug by modifying its buffer chemical properties. For example, using chemically modified DNA backbones like phosphorothioate or a G-quadruplex sequence (Fig. 5A, left panel), we designed various doxorubicin DNA

buffers with increased resistance to chemical degradation in mice serum that contains various nucleases (supplementary Fig. 12)[38]. We first showed that these chemical modifications do not significantly affect the binding affinity for doxorubicin since the $K_D$ varied from 1 to 3 μM in mouse serum (supplementary Tables 1, 2). Using HPLC-fluorescence measurements, we also validated that doxorubicin alone was mostly chemically degraded after one hour when exposed to mice serum (Fig. 5A, middle panel, black line). We then showed that the doxorubicin half-life in mouse serum increases proportionally with the buffer stability (Fig. 5A, middle and right panel). For example, the phosphorothioate buffer variant, with increased serum stability, $t_{1/2} = 10$ vs 6 h of the unmodified buffer, led only to 55% of doxorubicin degradation after 36 h in serum compared to 85 and 90% when employing the unmodified D0 or no buffer, respectively (Fig. 5A, middle panel). Molecular buffers can also be employed to modify the drug-like properties, such as the hydrophilic/hydrophobic balance. To demonstrate this feature, we measured the partition coefficient of doxorubicin between octanol and water: log $P$ (Fig. 5B, left panel). Free doxorubicin and the different formulations were first solubilized in PBS and delicately added upon an octanol layer at a ratio of 1:1 v/v. Free doxorubicin rapidly shifted to the more hydrophobic octanol phase with log $P = 0.75$). In contrast, the 0.5-D0 formulation containing the hydrophilic buffer retained more doxorubicin in the aqueous phase with log $P = 0.1$) for a longer period of time (Fig. 5B, middle and right panel). Increasing the hydrophobicity of the buffer through covalent conjugation of C16 or cholesterol moieties to the buffer resulted in the further increase of apparent log $P$ of doxorubicin from 0.1 to 0.45 (Fig. 5B right panel).

### In vivo proof of concept of molecular buffers

An important challenge with drug dosage is that each drug may undergo different biodistributions and distribution volumes in the body, which are often not optimal[39]. When injecting 10 mg/kg of doxorubicin in mice, for example, less than 5% of this dose remains in the blood circulation after only 5 min (Fig. 5C left and middle panel). To program doxorubicin distribution through the body and, for example, to sequester it in the blood, we employed the hydrophilic buffer D0-L-

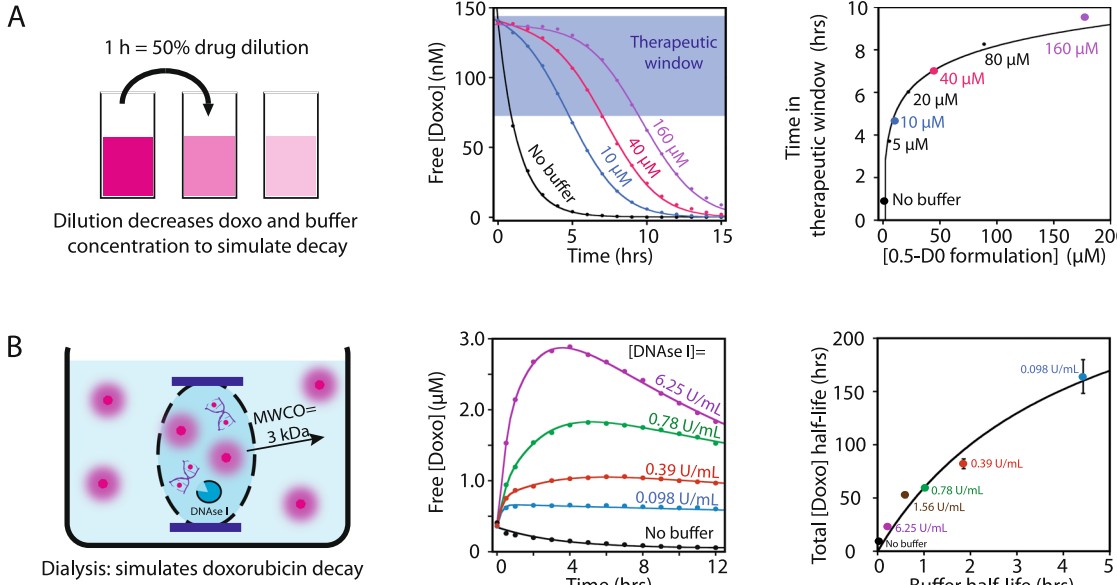

**Fig. 4 | Programming the doxorubicin pharmacokinetic profile using molecular buffers.** (**A**, left panel) Simulating doxorubicin and buffer degradation using serial dilutions. (**A**, middle and right panel) Large reservoirs of drug/buffer formulation of 0.5-D0 maintain the free doxorubicin concentration in an arbitrarily selected therapeutic window depicted by the blue square between 75 and 150 nM for a much longer period despite many dilution cycles (for middle panel, black line: no buffer; blue line: 10 μM reservoir; pink line: 40 μM reservoir and purple line: 100 μM reservoir). Data in the middle panel are fitted with equation E11 and data in the right panel with equation E14 (Supplementary Information). (**B**, left panel) Simulating doxorubicin clearance using a dialysis cassette and buffer degradation using DNAse I nuclease activity with formulation 0.5-D0. (**B**, middle panel) The pharmacokinetic

profile of doxorubicin (10 μM) can be shaped by modifying the stability of its DNA buffer (D0) using various concentrations of DNAse I (black line: without DNAse I; blue line: 0.098 U/mL; red line: 0.39 U/mL; green line: 0.78 U/mL; purple line: 6.25 U/mL and pink line: 100 U/mL). Data were collected for 24 h and fitted with equation E18−see Supplementary Information. (**B**, right panel) The doxorubicin half-life can be increased by increasing the buffer half-life (data fitted with a Michaelis−Menten equation). For right panels **A/B**: each data point is derived from the fitting of a single doxorubicin release kinetic from middle panels A/B and all error bars were derived from the same fitting. Data were presented as mean values ± SD. All data were obtained using fluorescence measurements.

thioate, which is a longer D0 buffer variant with six doxorubicin binding sites (supplementary Table 2), which itself displays a much higher half-life than the drug with $t_{1/2} = 22$ min vs 1 min (supplementary Fig. 13). When injecting doxorubicin using a 0.9-D0-L-thioate formulation, the apparent $t_{1/2}$ of total doxorubicin in blood circulation increased up to 4.4-fold (blue data, Fig. 5C middle and right panel). To further increase the doxorubicin half-life, we have grafted a C16 fatty acid onto the D0-L-thioate buffer to promote binding to albumin protein (supplementary Figs. 14, 15). This buffer modification increased the $t_{1/2}$ of the molecular buffer by tenfold, which increased the $t_{1/2}$ from 22 min to 4 h (supplementary Fig. 13), which in turn, increased the $t_{1/2}$ of doxorubicin by 18-fold, up to 19 min (Fig. 5C, right panel). When injected in a 0.9-D0-L-thioate-C16 formulation, the total concentration of doxorubicin in circulation remained higher than 35 μM, even after 2 h (Fig. 5C middle panel). These results highlight that in addition to maintain free doxorubicin concentration in a self-regulated manner, molecular buffers can also be programmed to increase drug half-life in vivo.

We further explored how molecular buffers control doxorubicin biodistribution in CD-1 live mice by performing a bioimaging analysis of free doxorubicin fluorescence in seven internal organs. Five minutes after intravenous injection of free doxorubicin at 10 mg/kg, 95% of free doxorubicin had left the blood circulation (Fig. 5 middle panel), and all organs displayed high levels of doxorubicin fluorescence, especially the liver, the heart and the kidneys (Fig. 6A, B and supplementary Figs. 16, 17). After 2 h, no doxorubicin was left in the blood circulation (Fig. 6C, D), and the average doxorubicin fluorescence in all organs had already dropped by 35% (supplementary Fig. 17). In contrast with when injected with molecular buffers, most doxorubicin remained in circulation after 5 min: 65% for D0-L-thioate and 99% for D0-L-thioate-C16 (Fig. 6B). Even after 2 h, doxorubicin was still present in blood

circulation at 1% for D0-L-thioate and 7% for D0-L-thioate-C16 (Fig. 6D). Additionally, we noticed that the buffers prevented a sharp spike in drug concentration (or burst effect) in most organs a few minutes after injection. For example, the average doxorubicin fluorescence in organs after 5 min was decreased by 45% when employing D0-L-thioate and by 63% when employing the D0-L-thioate-C16 (Fig. 6A, B and supplementary Figs. 17−19). These results highlight the programmable feature of molecular buffers that can be attached to specific biomolecules to optimize drug distribution. For example the D0-L-thioate being attached to albumin via a C16 anchor allows to maintain doxorubicin in blood circulation, while minimizing its biodistribution elsewhere.

We then assessed how the changes in biodistribution of organs vs. plasma and pharmacokinetics of immediate vs. sustained release of doxorubicin-induced by the buffers affect its toxicity. In agreement with the toxicity of doxorubicin, we first observed that mice injected with doxorubicin gained weight more slowly than control mice: only 10% of the regular weight gain within two weeks (Fig. 6E)[40]. In contrast, when treated with our buffer formulations, mice gained 70% for D0 and 80% for D0-C16 of the control weight gain. Surface electrocardiograms, ECG, recorded before and two weeks after injection showed that doxorubicin alone had no effect on ECG parameters. However, heart rates were specifically increased in mice treated with the 0.9-D0-L-thioate (92 ± 20 bbp) and 0.9-D0-L-thioate-C16 (84 ± 13 bbp) formulations (see Fig. 6F and supplementary Table 3). ECG analysis, however, showed that the two buffers alone did not affect the heart rate or any other ECG parameters significantly. We hypothesize that the distinct physiological outcome of the mice treated with the molecular buffer formulation: an increase in heartbeat with no weight loss may be attributable to the longer half-life of doxorubicin in blood and its reduced biodistribution elsewhere.

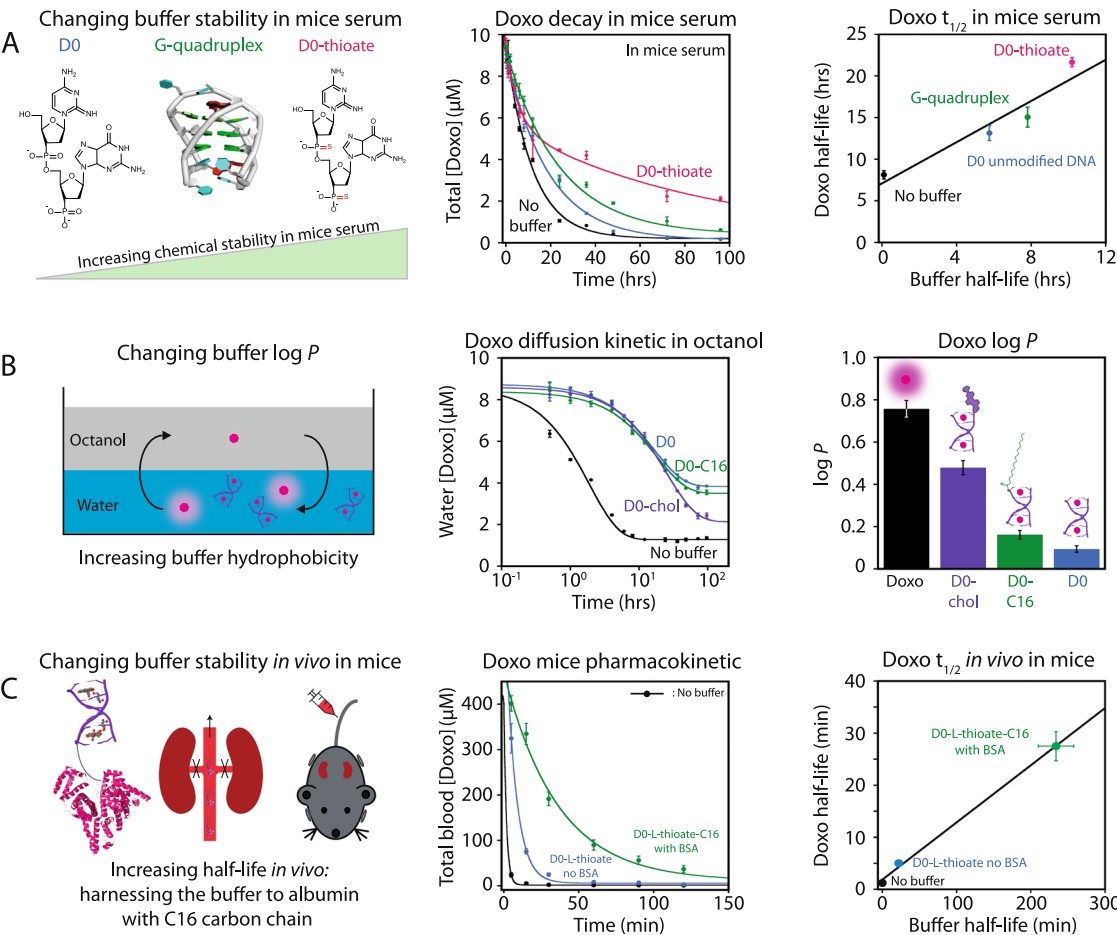

**Fig. 5 | Optimizing the pharmacological properties of doxorubicin through chemical modifications of its buffer. A** Increasing the chemical stability of doxorubicin in mouse serum by increasing its buffer chemical stability (left panel) resulted in increasing the doxorubicin half-life (middle and right panels). Black line: no buffer; blue line: D0 buffer; green line: G-quad buffer; pink line: D0-thioate buffer. Total doxorubicin concentrations were determined using HPLC-fluorescence measurements, while DNA buffer half-lives were determined using SYBR green fluorescence (supplementary Fig. 12). Data in the middle panel were fitted with equation E19—see Supplementary Information) and a linear regression was used for the right panel. For each condition, $n = 3$ independent experiments. Data were presented as mean values ± SD. **B** Programming the doxorubicin partition coefficient log *P* (left panel) by modifying the hydrophobicity of the buffer resulted in different water-to-octanol diffusion (middle panel), which changed the

doxorubicin log *P* value (right panel). Black line: no buffer; blue line: D0 unmodified buffer; green line: D0-C16 (D0 with C16 moiety); purple line: D0-chol (D0 with cholesterol moiety). Data in the middle panel are fitted with equation E19—see Supplementary Information. Data and error bars from the right panel are derived from the fitting of the middle panel. For each condition, $n = 3$ independent experiments. Data were presented as mean values ± SD. **C** Increasing the doxorubicin blood circulation time using molecular buffers (left panel) in IV-injected CD-1 mice. Both molecular buffer D0-L-thioate (blue line) and D0-L-thioate-C16 (green line) increased doxorubicin half-life in vivo (middle and right panel) compared to non-buffer (black line). Data in the middle panel fitted with equation E19—see Supplementary Information. Data and error bars from the right panel are derived from the fitting of the middle panel. For all conditions, $n = 6$ and data are presented as mean values ± SD.

We also performed a histopathological analysis of organs, which revealed a classical sign of doxorubicin cardiotoxicity, cardiomyocyte vacuolization, in one of the six mice treated with doxorubicin in the absence of buffers (F1—see Fig. 6G)[41]. Mice treated with the buffer formulations did not show any specific sign of degeneration. Finally, we also analyzed various toxicity biomarkers two weeks after the doxorubicin single injection. No significant difference between formulations was observed for heart and kidney biomarkers (supplementary Fig. 20 and supplementary Table 4). However, an increase in the ALT liver enzyme in plasma was detected in mice treated with the 0.9-D0-L-thioate and the 0.9-D0-L-thioate-C16 formulations (supplementary Fig. 20A). Taken together, these findings demonstrate that the precise/sustained pharmacokinetics of doxorubicin produced by our self-regulated buffers reduce some physiological effects of doxorubicin like weight loss and cardiomyocytes vacuolation, while enhancing some others like heart rate.

With this in vivo proof of concept showcasing its potential, we believe that molecular buffers could improve the delivery of drugs that

display a small therapeutic window and/or for which selection of an optimal therapeutic dosage remains challenging. We also believe that molecular buffers could further contribute to optimize drug properties such as its log *P* (optimal between −0.4 to 5[42]) its biodistribution, as well as potentially reduce drug resistance by maintaining drug concentration in the target range. To facilitate the development of future molecular buffers, we propose the following short guideline: (1) Find a known binding partner for the drug of interest. It could be found in the scientific literature or in a database (e.g., aptagen for aptamers and peptides) or alternatively, selected via various techniques (e.g., SELEX, phage-display…); (2) Measure the binding affinity ($K_D$) and stoichiometry between the molecular buffer and the drug and optimize its $K_D$ to be near the target concentration. This can be realized via various strategies[43,44] that include point mutations[45], creating a structure-switching molecular buffer and modifying its switch equilibrium via mutation[24], inhibitors or activators[45–47]; (3) Validate that drug binding to its molecular buffer inactivates its therapeutic activity through cell culture viability tests; (4) Modify the molecular

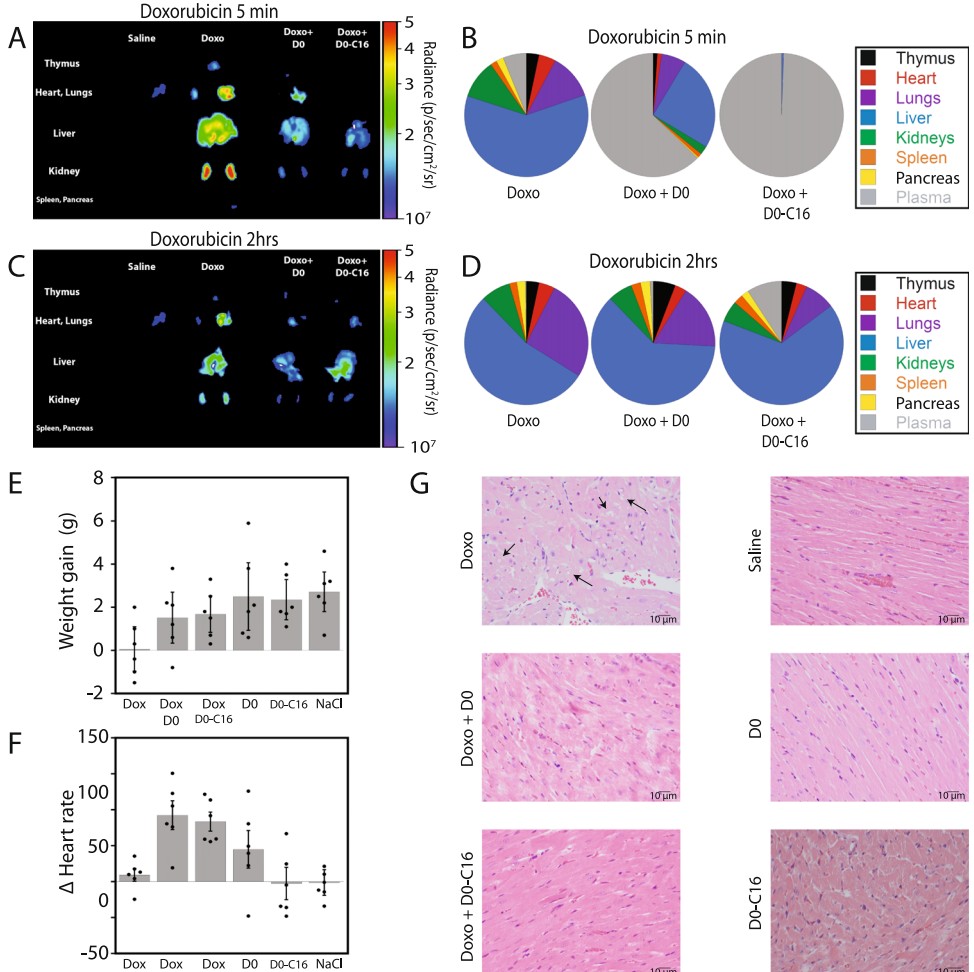

**Fig. 6 | Molecular buffers enable the programming of the biodistribution of a drug. A–D** Ex vivo bioimaging of 10 mg/kg doxorubicin formulations in IV-injected CD-1 female mice with $n = 3$. The formulations 0.9-D0-L-thioate (see D0 in the figure) and 0.9-D0-L-thioate-C16 (see D0-C16 in the figure) were programmed to maintain a free concentration of doxorubicin of 5 μM. The percentage of doxorubicin in each organ/serum was determined using fluorescence imaging for the organs and HPLC for the serum. **E**, **F** Weight gains and changes in heart rate in CD-1 female mice two weeks after IV injection of 0.9-D0-L-thioate and 0.9-D0-L-thioate-C16 formulations. For each condition in both panels with $n = 6$ and data are presented as mean values ± SD. **G** Histopathological analysis of cardiac tissue after 2 weeks shows degenerative cardiomyocyte vacuolization in one mouse when doxorubicin was injected without a DNA buffer. For each, $n = 6$ mice were analyzed with 2 sections each ($n = 12$ in total). Note: for simplicity in the figure, D0 stands for D0-L-thioate and D0-C16 stands for D0-L-thioate-C16.

buffer structure to optimize its chemical stability, biodistribution, and log $P$. For example, we have changed the DNA backbone of our molecular buffer to phosphorothioate to reduce its degradation from nuclease and have also added a C16 anchor aimed to bind to albumin that maintains the buffer and doxorubicin in blood circulation. PEG modifications could also improve the buffer half-life by reducing its clearance rate[48]. These modifications should not significantly affect the drug binding affinity ($K_D$). Re-optimize $K_D$ if needed; (5) Characterize the buffer and drug/buffer formulation pharmacokinetics in vivo to determine the efficiency of the molecular buffer; (6) Select an optimal formulation (drug/buffer ratio) and dosage to optimize treatment outcome.

## Discussion

In this paper, we have demonstrated how we can engineer bio-inspired self-regulated molecular buffers that are programmed to release and maintain a precise drug concentration in vivo. We have taken advantage of the programmable nature of DNA chemistry to develop these buffers for the chemotherapeutic drug doxorubicin and the antimalarial drug quinine. We showed that these buffers can be readily programmed to maintain a targeted drug concentration, that they can

be employed as a drug reservoir to prolong the circulation time of a drug and that their stability can be tuned to match the desired pharmacokinetics profile. We also showed how these buffers can modify drug-like properties like log $P$ and blood circulation time. Finally, we showed how these buffers can be programmed to optimize the biodistribution of a drug and its circulation time and minimize its burst phase minutes after injection, thus reducing the side effects of the drug-like cardiotoxicity and weight loss.

The concept of a molecular buffer is universal and could be envisaged with any type of drug using any type of chemical receptor-like DNA or proteins, provided that the molecular buffer has a specific affinity for the drug of interest. As our proof of concept buffer, we have employed nucleic acid chemistry given its high convenience of easy synthesis, modification, and programmability, its biocompatibility and its relative universality to bind any molecular target[49–51]. DNA molecular buffers can also be further incorporated into almost any commonly used passive drug carrier like lipid nanoparticles[52] or liposomes with DOXIL[27] to achieve both optimal self-regulated delivery and maximized molecular buffer half-life. Although our proof of concept buffer utilized DNA, we believe that any other biocompatible material with a high affinity for a specific drug, including proteins, peptides,

sugars, or lipids, could be adapted and designed in an efficient programmable molecular buffer.

In addition to precisely control the concentration of a drug over time in vivo, molecular buffers can also be exploited to deliver drugs more efficiently to targeted organs. Here, we described how the attachment of a C16 moiety to the buffer, thanks to albumin binding, sequesters the drug-loaded buffer in the blood circulation, extending the doxorubicin half-life by 18-fold. Alternatively, specific targeting moieties, such as small molecules[23], aptamers[50,51] and peptides[53,54], could guide the buffers to specific organs or cell types like tumors. The buffer could also be employed with drugs that are poorly internalized by the cell, since various modifications could alter the cellular trafficking of the buffer and its bound drug. For example, cell-penetrating peptides increase oligonucleotide uptake by the cell[55] and other modifications can target specific cellular pathways like the furin or transferrin pathways for example[56]. The main advantage of this approach over drug conjugates[57–59] is that molecular buffers can be readily modified without affecting the molecular structure of the drug and, therefore, its specificity and selectivity. Overall, we believe that by programming the concentration and the biodistribution of drugs, molecular buffers can potentially enhance the efficiency/specificity of any drug while minimizing its side effects. As we enter the era of smart DDS, we envisage that molecular buffers will provide another generation of smart therapeutic tools that take into account the individual pharmacokinetic profiles of patients and help prevent medical overdosing and medication errors.

## Methods

All graph plotting and data fitting were performed with Kaleida Grap 4.0 (Synergy software PA, USA).

### Oligonucleotide synthesis and purification

Oligonucleotides were synthesized on an H-6 DNA synthesizer from K&A Laborgeraete (Schaafheim, DE). Standard reagents for DNA synthesis with phosphoramidite chemistry were purchased from ChemGenes (Wilmington, MA). Sulfurizing reagent (DDTT) and fast deprotecting phosphoramidites from ChemGenes (Wilmington, MA) were used for the synthesis of phosphorothioate oligonucleotides. All oligonucleotides were synthesized from 3′ to 5′ terminal end starting with controlled pore glass (CPG) beads from Biosearch Technologies (Petaluma, CA). A palmitic acid (C16) CPG was used to prepare D0-C16 oligonucleotide, while a 5′ cholesterol phoroamidite was used to incorporate a cholesterol moiety in the 5′ end for D0-chol oligonucleotide. Following DNA synthesis, oligonucleotides were cleaved from a solid support and deprotected with 1 mL of 30% aqueous ammonium hydroxide at room temperature for 36 h. For phosphorothioate oligonucleotides, cleavage and deprotection were performed with a 1:1 mix of aqueous methylamine and 30% aqueous ammonium hydroxide for 2 h at room temperature. After deprotection and cleavage, oligonucleotides were purified on a P-8 solid-phase extraction purifier from K&A Laborgeraete (Schaafheim, DE) with MicroPure II columns from Biosearch Technologies (Petaluma, CA). Oligonucleotides were eluted with 1 mL of 50% acetonitrile in water and recovered after evaporation of the elution solution in a SpeedVac (Fisher Scientific, Waltham MA) at 60 °C for 3 h. Oligonucleotides were dissolved in 100 μL of ultrapure water from an EMD Millipore Milli-Q system (Billerica, MA) with 18.2 MΩ cm resistivity to yield a concentration ≥1 mM. Oligonucleotides were quantified by UV absorption at 260 nm with sequence-specific extinction coefficients calculated using the IDT biophysics webserver. Oligonucleotide sequences are reported in supplementary Table 1.

### Fluorescence spectrophotometer measurements

All fluorescence measurements were acquired on a Cary Eclipse Fluorescence Spectrophotometer from Agilent Technologies (Santa Clara, CA). Quartz cuvettes were used for all fluorescence measurements except with mouse serum, for which plastic disposable cuvettes were used. For each experiment, either scan mode or kinetic mode was used.

**Fluorescence scan.** Cary WinFLR Scan Software Version: 1.2(147) (Agilent, Santa Clara, CA, USA) was used. For doxorubicin fluorescence measurements, the excitation wavelength was set at 480 nm, and the emission scan ranged from 595 to 596 nm, with an averaging time of 10 s. The photomultiplier detector was set to 800 V with a 5-nm slit for excitation and emission. For quinine, the excitation wavelength was set at 330 nm, and the emission scan ranged from 390 to 391 nm, with an averaging time of 10 s. The photomultiplier detector was set to 650 V with a 5-nm slit for excitation and emission. Scans were recorded three times and averaged for the fluorescence value at 595 nm for doxorubicin or 390 nm for quinine.

**Binding and stoichiometry curves.** Binding curves were obtained by adding increasing amounts of DNA aptamer to doxorubicin (BioShop Canada, Inc., Burlington, ON) or quinine (Sigma Aldrich, Oakville, ON) and measuring the resulting fluorescence quenching. Scan mode was used, and the temperature was set at 20 °C. The buffer solution used for doxorubicin was 50 mM $Na_2HPO_4$ and 100 mM NaCl at pH 7.00 (except for the G-quadruplex sequence, for which sodium was replaced with $K_2HPO_4$ and KCl). The buffer solution used for quinine was 50 mM $NaH_2PO_4$, and the pH was increased to 7.00 with NaOH (50 mM).

For the binding curve with albumin, fluorescein-labeled DNA was used, and the affinity was measured with fluorescein fluorescence quenching upon albumin addition.

Stoichiometry was measured with a binding curve using doxorubicin or quinine concentrations above the $K_D$. Scan mode was used, and the temperature was set at 20 °C. The same buffer solution from binding curves was used.

**Buffer titration curves.** Various DNA aptamer solutions at similar concentrations were titrated with increasing amounts of doxorubicin or quinine. Fluorescence was recorded for every drug increment and converted to free concentration with a fluorescence standard curve after correction with the quenching efficiency. Scan mode was used, and the temperature was set at 20 °C. The same buffer solution from binding curves was used.

**Dialysis kinetics.** A 3-mL Slide-A-Lyzer dialysis cassette with MWCO = 3 kDa from Thermo Fisher Scientific (Waltham, MA) was used. Dialysis cassettes containing doxorubicin and DNA aptamer (MW = 7.3 kDa) were immersed in a beaker containing 1 L of DNAse I buffer solution (10 mM Tris, 2.5 mM $MgCl_2$, 0.5 mM $CaCl_2$ at pH 7.60). At $t = 0$, DNAse I was added to the dialysis cassette. For every time-point of the kinetics, 800 μL of the dialysis cassette was drawn for fluorescence measurements and quickly poured back in the cassette to continue dialysis. Fluorescence values were converted to doxorubicin concentrations with a standard curve. Scan mode was used, and the temperature was set at 37 °C. The same buffer solution from enzymatic assays was used.

**Fluorescence kinetic.** Cary WinFLR Kinetics Software Version: 1.2(146) (Agilent, Santa Clara, CA, USA) was used. For doxorubicin fluorescence measurements, the excitation wavelength was set at 480 nm, and the emission wavelength was set at 595 nm, with averaging times of 0.1 s for fast events and 3 s for slower events. For doxorubicin, the photomultiplier detector was set to a voltage of 800 V with a slit of 5 nm for excitation and emission. For quinine fluorescence measurements, the excitation wavelength was set at 330 nm, and the emission wavelength was set at 390 nm, with the same averaging times as doxorubicin and

with a voltage of 650 V for the photomultiplier detector and a slit of 5 nm for excitation and emission. An oil layer (300 μL) was added for kinetic experiments lasting more than 1 hour to prevent evaporation.

**DNASe assays.** Doxorubicin and its DNA aptamer were incubated with various concentrations of DNASe I from New England Biolabs (Ipswich, MA). Doxorubicin fluorescence was measured over time and converted to doxorubicin concentration as previously described. The kinetic mode was used, and the temperature was set at 37 °C. All experiments with DNASe I were performed with buffer solution provided by New England Biolabs (Ipswich, MA): 10 mM Tris, 2.5 mM MgCl$_2$, 0.5 mM CaCl$_2$ at pH 7.60. Binding affinity and stoichiometry in DNASe I buffer solution were measured as previously described.

**DNA degradation in mouse serum.** DNA degradation in mouse serum was monitored with fluorescence at 37 °C using SYBR green reporter dye (concentration excess of 50X) from Sigma Aldrich (Oakville, ON). The kinetic mode was used, and the temperature was set to 37 °C. Non-Swiss Albino Mouse Serum from Innovative Research, Inc. (Novi, MI) was used. Binding affinity and stoichiometry in mouse serum were measured as previously described.

## Stopped-flow fluorescence measurements
All fluorescence measurements were acquired on an SX20 stopped-flow spectrophotometer from Applied Photophysics (Leatherhead, UK) with excitation wavelengths set to 480 nm and a 9.3-nm bandwidth. Fluorescence emission was measured by reading fluorescence intensity using a high-pass glass filter with 495-nm cutoffs and a photomultiplier set to 400 V. The injection system consisted of two syringes: 2.5 and 0.5 mL. The temperature was set at 37 °C with a water bath, and the same buffer solution from enzymatic assays was used. Chirascan v.4.2.15 (Applied Photophysics, Leatherhead, UK).

**Dissociation kinetics.** Doxorubicin and DNA aptamer were placed in a 0.5-mL syringe, and a buffer solution was added to the 2.5-mL syringe. Injection in the stopped-flow spectrophotometer yielded a 1/6 dilution of the doxorubicin/DNA aptamer solution, which resulted in doxorubicin dissociation. Fluorescence data were normalized after background correction with the maximum fluorescence (the fluorescence reached at the steady state) and minimal fluorescence (when no dilution occurred). To measure minimal fluorescence, the buffer solution in the 2.5-mL syringe was replaced with doxorubicin/DNA aptamer, and the recorded fluorescence was divided by 6 (at mixing, doxorubicin fluorescence is divided by 6, and then the bound fraction releases doxorubicin until equilibrium is reached).

**Binding kinetics.** Doxorubicin solutions were added to the 2.5-mL syringe and DNA aptamer solution to the 0.5-mL syringe. Data normalization after background correction was carried out with doxorubicin maximum fluorescence (measured when the DNA aptamer was replaced with buffer solution in the injection syringes).

## HPLC-fluorescence measurements
**HPLC-fluorescence quantification.** Doxorubicin and fluorescein concentrations were determined using fluorescence. All HPLC analyses were performed on a 1260 Infinity II LC System from Agilent Technologies (Santa Clara, CA) equipped with an XBridge Oligonucleotide BEH C18 column from Waters (Milford, MA). The temperature was set at 40 °C, and the injection volume was 22.5 μL. TEAA mobile phase (100 mM triethylamine/acetic acid, pH = 7.00) (Fisher Scientific, Waltham MA) was used for ion-pairing binding to the column, and 100% HPLC grade acetonitrile was used for elution. The UV detector was set to measure absorbances at 260 and 480 nm. The fluorescence detector was set with an excitation wavelength of 480 nm and multiple emissions with wavelengths of 520, 550, 595, and 650 nm and a PMT

gain of 18. To determine the doxorubicin or fluorescein fluorescence from each chromatogram, the AUC was measured with the software Chem Station provided by Agilent Technologies (Santa Clara, CA). A standard curve of doxorubicin and fluorescing fluorescence was obtained before sample analysis. Samples were run in triplicate. Fluorescence AUC was converted to concentration with the standard curve and plotted through time (for serum or plasma samples, standard curves were acquired with serum or plasma spiked with the relevant analyte).

**Water/octanol partition of doxorubicin.** Doxorubicin with or without aptamer buffer in PBS (0.9 mL) was transferred in a 2-mL flat-bottom Eppendorf tube, and a 2 × 7-mm magnetic stirrer (VWR item 47751-506, Radnor, PA) was added. A layer (0.9 mL) of octanol (equilibrated overnight with PBS) was added on top of the aqueous phase, and the tube was placed in a 37 °C temperature-controlled water bath with 120 rpm shaking. For each time-point, 40 μL of the aqueous phase was removed carefully with a gel-loading pipet tip and transferred to an HPLC loading tube for a fluorescence quantification of the doxorubicin content (each quantification was performed in triplicate). No extraction was required for these samples (as they were already in PBS). An equal volume of the octanol phase was removed and discarded to maintain the same 1:1 volume ratio between the aqueous and octanol phases. The fluorescence decrease from the aqueous phase (diffusing in the octanol phase) was quantified over time with HPLC.

**Extraction of doxorubicin from mouse serum and plasma.** Doxorubicin was extracted from mouse serum or plasma before HPLC analysis. To do so, 10 μL of mouse serum or plasma containing doxorubicin was mixed with 1 μL of 100 μM daunorubicin (internal standard). Then, 100 μL of HPLC grade acetonitrile (Sigma Aldrich, Oakville, ON) was added and mixed thoroughly for 30 s to precipitate proteins. The mixture was centrifuged for 10 min at 10,000×g at 4 °C, and the supernatant was recovered. The centrifugation procedure was repeated twice. The acetonitrile was evaporated in a SpeedVac at 35 °C for 30 min. The recovered solid was dissolved in 25 μL of mobile phase (Fisher Scientific, Waltham MA). Only plasma samples containing doxorubicin (formulation F1, F2, and F3) were analyzed for the in vivo pharmacokinetic experiment.

**DNA samples prepared from mouse plasma.** For sample preparation, 10 μL of mouse serum or plasma containing fluorescein-labeled oligonucleotides were mixed with 1 μL of 100 μM internal standard and diluted with 90 μL of the mobile phase. Two internal standards were used for DNA fluorescein fluorescence quantification: D0-thioate-FAM-C12 (standard for D0-thioate-FAM-C16) and D0-FAM (standard for D0-thioate-FAM). The solution was injected directly into the HPLC (injection volume of 90 μL). Fluorescence quantification was performed as described previously.

## Molecular buffer application in vitro
**Cell culture.** HeLa and HCT116 cell lines were obtained from ATCC (Manassas, USA). Both cell lines were authentified based on morphology and PCR assays with human-specific primers. Both cell lines were negative for mycoplasma. HeLa cells were cultured in DMEM 319-005-CL from Wisent Bioproducts (St-Bruno, QC) supplemented with 10% fetal bovine serum (FBS). HCT116 cells were cultured in DMEM 319-005-CL from Wisent Bioproducts (St-Bruno, QC) supplemented with 10% fetal bovine serum (FBS). Cell cultures were incubated at 37 °C with 5% CO$_2$.

**Toxicity assay.** Cells were seeded in a 96-well plate at a density of $5 \times 10^3$ cells per well. After 24 h, the culture medium was removed, and a fresh medium with various concentrations of doxorubicin or doxorubicin-aptamer was added. After 48 h of incubation, the medium

was removed, and 100 μL of DMEM 319-050-CL from Wisent Bioproducts (St-Bruno, QC) supplemented with 10% FBS was added to each well with 20 μL of 440 μM resazurin (Sigma Aldrich, Oakville, ON) in DPBS 1X. Cells were incubated an additional 4 h. Fluorescence was measured at 590 nm with a 560-nm excitation wavelength to determine the metabolic reduction of resazurin with a Gemini™ XPS Microplate Reader with software SoftMax® Pro GxP v7 (Molecular Devices, San Jose CA). Each condition was repeated eight times to ensure reproducibility. The binding affinity and stoichiometry of doxorubicin-aptamer in a culture medium at 37 °C were measured as previously described.

**Confocal microscopy.** μ-Slide 8-well chambered coverslips (ibidi, Madison, WI) were used. Before seeding, the coverslips were treated with 15 μg/mL poly-lysine solution (300 μL/well) (diluted from 0.1% poly-lysine solution of Sigma Aldrich, Oakville, ON) for 30 min at 37 °C. Cells were seeded at a density of $15 \times 10^3$ cells per well and incubated for 24 h. Subsequently, the culture medium was removed, and a fresh medium with various concentrations of doxorubicin or doxorubicin-aptamer or fluorescent buffer (D0-FAM) was added (two wells per conditions). Cells were incubated for 1 h at 37 °C and rinsed with DPBS 1X. Cells were fixed with fresh 4% formaldehyde solution (diluted in pure water from a fresh 37% stock from Sigma Aldrich, Oakville, ON) for 15 min and washed again with DPBS 1X. For colocalization, 300 nM DAPI solution (Sigma Aldrich, Oakville, ON) was added to each well and incubated in the dark for 5 min. Cells were rinsed thrice with DPBS 1X and mounted with four drops of ibidi Mounting Medium (ibidi, Madison, WI) for the confocal laser scanning microscopy (CLSM) observations. CLSM images were obtained on an LSM 700 inverted microscope imaging system (Zeiss, Oberkochen DE) with a 488-nm laser for doxorubicin (300–483 nm for emission) and a 405-nm laser for DAPI (544–800 nm for emission). Differential interference contrast microscopy (DIC) images were also recorded. Images were acquired and analyzed by Zeiss blue edition v2.3 software (Carl Zeiss Microscopy GmbH, Jena, Germany) and freeware ImageJ v1.51n for fluorescence quantification[60].

**Flow cytometry.** Cells were seeded in a 24-well plate at a density of $20 \times 10^3$ cells per well and incubated for 24 h. After seeding, the culture medium was removed, and a fresh medium with various concentrations of doxorubicin or doxorubicin-aptamer or florescent buffer (D0-FAM) was added (two wells per conditions). Cells were incubated for 1 h at 37 °C and rinsed with DPBS 1X. Cells were trypsinized and suspended in a FACS buffer (DPBS with 1% FBS and 0.1% sodium azide (Sigma Aldrich, Oakville, ON)). FACS analysis was performed on a FACScalibur (BD Sciences, Franklin Lakes, NJ, USA) with BD FACStation™ 6.1 Software (BD Sciences, Franklin Lakes, NJ). Fluorescence measurements were recorded with a linear scaling on an FL1-A fluorescence channel for fluorescein or an FL2-A channel for doxorubicin. Each measurement consisted of 20,000 events and was repeated three times. Data were analyzed with software FlowPy v5.2 (Department of Biosciences and Bioengineering, Indian Institute of Technology Guwahati, India).

**Electrophoresis**

All electrophoresis analyses were performed on an agarose electrophoresis system from Bio-Rad (Hercules, CA). Gels with 3% agarose were run in TAE buffer at 10 V/cm. Samples were prepared by adding the corresponding amount of fluorescent DNA in either 50% glycerol/water, mouse serum or 500 μM albumin solution in PBS (no loading dye added). After 75 min of migration, gels were visualized with a ChemiDoc XRS+ (Bio-Rad, Hercules CA) with a fluorescein filter for nucleic acids. To measure binding to serum albumin, D0-PS-FAM-C16 oligonucleotide was used, and the control oligonucleotide was D0-PS-FAM. ImageLab software 6.0.0 build 25 (Bio-Rad

laboratories, Saint-Laurent, QC, Canada) was used for data collection and analysis.

**In vivo mouse experiments**

Animals were housed in a pathogen-free environment with strictly controlled environmental conditions according to protocol #18-017 approved by the Université de Montréal Institutional Animal Care Committee (CDEA). They were housed inside an SPF (specific-pathogen-free) animal facility, exempted from the majority of known pathogens for murine species. Animals were housed in IVC (individually ventilated cages) and maintained on sterile (irradiated) mouse chow, sterile water (RO) and sterile (autoclaved) bedding and cages. Animals were maintained in a controlled microclimate (average room temperature was 22 °C and humidity between 40 and 50%). Animals were also maintained on a controlled light cycle of 12 h-12 h (12 h of light and 12 h of dark). Appropriate environmental enrichment was provided at any time as per CCAC (Canadian Council on Animal Care) current recommendations.

Normal 8-week-old female CD-1 Elite mice were ordered from Charles River and were acclimated 1 week before transfer in individual cages 3 days prior to starting experiments (day 1). No diet restrictions were applied, but a humid food formulation (DietGel® 76 A - ClearH2O, INC., Portland, ME, USA) was used to prevent. mouse dehydration during experiments.

**Pharmacokinetics.** For all conditions, $n = 6$. On day 1, 5 mL/kg tail vein injections were performed with six formulations:

F1: 10 mg/kg doxorubicin
F2: 10 mg/kg doxorubicin, 55 mg/kg D0-L-thioate oligonucleotide
F3: 10 mg/kg doxorubicin, 55 mg/kg D0-L-thioate-C16 oligonucleotide, 5% DMSO
F4: 55 mg/kg D0-L-thioate-FAM oligonucleotide
F5: 55 mg/kg D0-L-thioate-FAM-C16 oligonucleotide
F6: normal saline

A 10 mg/kg doxorubicin dose corresponds roughly to a 350 μM concentration of doxorubicin in blood circulation. The blood volume in a mouse is estimated to be 58.5 mL/kg according to NC3R^S. Blood samples (35 μL) were collected from the tail with a Minivette® POCT 50 μL K3EDTA (Sarstedt, Saint-Leonard QC), stored on ice for a maximum of 30 min and centrifuged 10 min at $2000 \times g$ at 4 °C. Blood samples were collected for eight time points: 5 min and 0.25, 0.5, 1, 1.5, 2, 4, and 24 h. Collected plasma samples were transferred into dry ice tubes and stored at −80 °C. All animals were sacrificed after 2 weeks, and organs were collected ex vivo for histopathological analysis.

**Ex vivo mouse organ bioimaging.** Mice were anesthetized with 3% isoflurane/$O_2$ and perfused intracardially with 10 mL PBS to rinse all blood from the circulation. Organs were then removed ex vivo, and fluorescence imaging was performed on an IVIS Spectrum In Vivo Imaging System with Living Image software v4.7.3 (PerkinElmer, Waltham MA). The GFP excitation passband was used (445–490 nm) and the DsRED emission passband was used (575–650 nm) for the fluorescence imaging. For each formulation, two time points were chosen (5 min and 2 h), with $n = 3$ for each time point. Images were processed with Living Image software.

**Electrocardiogram recordings.** Mice were anesthetized with 5% isoflurane/$O_2$ and maintained with 2% isoflurane/$O_2$. The lubricant was applied to the eyes to prevent the corneas from drying out. Mice were then placed on a heating pad, and the temperature was monitored with a rectal thermometer until it reached 37 °C. ECGs were recorded with a Powerlab data acquisition instrument (ADInstrument, Sydney, AU) with LabChart software v8.1.9 (ADInstruments, Sydney, Australia). Each electrode (four) was placed on a different leg, and the ECG was recorded with lead III.

**Histopathological analysis.** Organs were harvested ex vivo and directly transferred in pre-labeled Biopsy Cassettes (Simport Scientific, Saint-Mathieu-de-Beloeil, QC). The cassettes were transferred in jars containing 10% formalin solution (Chaptec, Inc., Montréal, QC) and were fixed for 48 h at room temperature. The following organs were harvested for fixation: heart, lungs, liver, kidneys, spleen, and intestine. The organs were processed at the Histology service from the Institute for Research in Immunology and Cancer (IRIC) and were stained with hematoxylin and eosin. The histopathological analysis of tissue slices was performed by Dr. Jeremie Berdugo from Hôpital Maisonneuve-Rosemont.

**Toxicological analysis.** For toxicological analysis, blood samples (1 mL) were collected by cardiac puncture with 29 G 1 cc syringes, transferred in S-Monovette® 1.2 mL EDTA K3 and centrifuged 10 min at $2000 \times g$ at 4 °C. Collected plasma samples were aliquoted at a volume of 100 μL in dry ice tubes and stored at −80 °C before analysis. Biomarkers analyzed were: troponin T, N-terminal prohormone of brain natriuretic peptide (NT-proBNP), urea, creatinine, total and conjugated bilirubin, alanine transaminase (ALT), aspartate transaminase (AST), alkaline phosphatase (ALP), and gamma-glutamyltransferase (GGT). Sample processing and analysis were performed automatically at the clinical biochemistry department of the Hôpital Maisonneuve-Rosemont.

### Reporting summary
Further information on research design is available in the Nature Research Reporting Summary linked to this article.

## Data availability
All data have been deposited on figshare: https://doi.org/10.6084/m9.figshare.20326638. Source data is available for Figs. 2–6 and Supplementary Figure 1, 3, 4, 5, 6, 9, 10, 11, 12, 13, 15–19, and 20 in the associated source data file. Source data are provided with this paper.

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

## Acknowledgements

This work was supported by the National Sciences and Engineering Research Council of Canada through grants RGPIN-2020-06975 (A.V.-B.) and RGPIN-06403 (A.V.-B.), the Canada Reserch Chairs, through grant 950-230012 (A.V.-B.), by Les Fonds de recherche du Québec—Nature et technologies through grant 2019-PR-256552 (A.V.-B.), and by Le regroupement québécois de recherche sur la fonction, l'ingénierie et les applications des protéines (PROTEO). A.V.-B. holds the Canada Research Chair in Bioengineering and Bionanotechnology, Tier II. A.D. is a Natural Sciences and Engineering Research Council fellow (Québec, Canada). We have special thanks for Manon Laprise and Dr. Ovidiu Jumanca and other personnel from the Montreal Clinical Research Institute (IRCM) Animal Facility for their help and assistance with mice experiments. We also thank Christian Charbonneau from the Institute for Research in Immunology and Cancer (IRIC) bioimaging platform. We would also like to thank Laurianne Pham and other members of the Vallée-Bélisle laboratory for their helpful discussion of the manuscript.

## Author contributions

A.D. and A.V-B. conceived and designed all experiments and equations. A.D. performed all experiments except FACS and confocal microscopy imagery (R.M.D. and J.L.C.); cell culture (S.H. and L.D.); histopathology slide analysis (J.B.); ECG data analysis (V.L. and C.F.); biomarkers quantifications (V.D.G.). D.L. provided assistance with data analysis and J.L.C. provided project oversight. A.D. and A.V.-B. created the figures and wrote the manuscript, which was then reviewed by all authors. A.V.-B. provided project oversight and funding.

## Competing interests

The authors declare no competing interests.
