## [Peer Review File · Nature Communications]

Programmable, self-regulated molecular buffers for precise,
sustained drug deliveryREVIEWER COMMENTS

Reviewer #1 (Remarks to the Author):

The manuscript "Programmable, self-regulated molecular buffers for precise, sustained drug delivery" by A. Desrosiers and co-authors presents an ingenious approach to controlling drug delivery in vitro and in vivo using DNA-based molecular buffers. By tuning the design of the molecular buffer (its drug affinity, stability in serum, ability to bind albumin, etc.) as well as its formulation (drug/buffer ratio) the authors have found a versatile strategy to control drug concentration ranges in variety of contexts for a multitude of potential drug types. As a proof-of-concept, the authors have performed a thorough characterization of their system from molecular design to in vivo assessments and have made intriguing findings regarding the effect of molecular buffers on drug properties. I heartily recommend this article for publication in Nature Communications.

In terms of items to review, the authors should consider answering the following questions and comments, which would help strengthen and clarify their claims with regards to novelty:

1. While the use DNA-based molecular buffers is very convincing to control the concentration of free drug in solution or plasma, it is unclear how it compares to other sustained drug delivery systems found in the literature. If possible, I recommend the authors provide more context in the introduction and conclusion on how their approach addresses challenges in the field of sustained drug delivery and what are the next challenges to address. This will help the reader appreciate the impact of their findings. Are there target values that indicate relevance for clinical applications? How helpful are molecular buffers towards achieving these targets? Currently, the introduction and conclusion are both very narrowly focused on their design and achievements and a lacking in explicit statements of the relevance to a broader context such as what would be expected in a clinical setting or comparisons against state-of-the-art formulations for sustained delivery, which complicates assessments of novelty and relevance.

2. Throughout the manuscript, the authors discuss how changing different aspects of the design and formulation of their molecular buffers result in modulation of the toxicity, reservoir quantity, stability, distribution and therapeutic/side effects of the drug released. Currently, the manuscript is very effective at showcasing the experiments that were performed and the conclusions that were drawn. However, many of these properties are tested separately in different contexts (in vitro vs in vivo, organ imaging vs in serum etc., some assays are done in solution with and without enzymes), which complicates the assessment of the key design principles that could be used in designing molecular buffer outside of the doxorubicin/quinine set explored. I recommend the authors provide key take-aways or explicit guidelines as a paragraph in their discussion for readers who would be interested in applying these molecular buffers to different contexts since they are presenting their strategy as a general platform. Is there an overall go-to recommendation for the Kd of the molecular buffer when the therapeutic window of a drug is known? Which modifications (e.g., phosphorothioate, G-quadruplex, cholesterol, C16, etc.) are more effective in promoting desired therapeutic outcomes over side effects in vivo? If these are not known, are there general recommendations for designing a DNA-based molecular buffer for a selected drug?

Minor edits suggested:

- I would recommend individual labeling of panels in figures instead of in groupings by themes to help connect with the main text. (e.g., when referring to Fig. 5B, there should be a title associated to panel explaining what is found specifically in this figure panel). Similarly, matching data point labels in related figures would be appreciated (e.g., Fig 4b and c, why is only the 160 uM data point highlighted? Fig 4e and f could also be matched similarly to facilitate figure reading).

Overall, the manuscript is excellent and well-written. It contains a sound methodology

with well-executed experiments. The authors use a clearly outlined approach and report outcomes in a systematic manner, enabling them to comment on important design parameters for assessing the performance of DNA-based molecular buffers in the context of sustained drug delivery.

Reviewer #2 (Remarks to the Author):

The study conducted by the authors is commendable as they are endeavouring to formulate self-regulated molecular buffers that would be able to regulate the desired concentration of the free drug both in vivo and in vivo. Albeit, there are a few concerns that needs resolution'

1) The authors claim that the molecular buffer is universal. The usage of the term universal can be slightly inaccurate as their formulation can never be parlayed into another system (another drug), as each new drug needs a unique interacting partner. If the authors still choose to cling on to the term universal, more clarifications are needed to justify the nature of universality of the buffer system

2) It would be great if the prowess of aptamer is accentuated by actually incorporating the term aptamer in the title. For example-'Programmable, self-regulated molecular buffer-assisted aptamer for precise, sustained drug delivery.'

3) Why did the authors use UREA-PAGE based purification to purify the oligos? UREA-PAGE based purification can be more selective as the synthesis of oligonucleotides using DNA synthesizer can result in multiple truncated forms along the way and this might affect the formulation of the self-regulated molecular buffers.

4) The authors mentioned 'A first approach consists of varying the KD of the buffer'. This is quite perplexing as to how one could change the KD of a buffer which is constant. How the authors were able to vary the KD of the buffer?

5) The authors also used the term 'drug degradation'. First of all, the authors ought to define the definition of the word degradation. Does it refer to the random decomposition of the drug due to any enzymatic or any action exerted by the system? Or does it refer to the dissociation of the drug from its receptor, in this case aptamers?

6) The authors claim that chemical modifications do not significantly affect the binding affinity for doxorubicin (KD varies from 1 μ M to 3 μ M) in mouse serum. How did the authors introduce G-quadruplex modification into the aptamers as the introduction of any G-rich sequence would definitely debilitate its binding affinity?

Reviewer #3 (Remarks to the Author):

I really enjoyed reading and reviewing the article by A.Desrosier and co-workers regarding a DNA drug delivery system displaying characteristics based on Le Chatelier's principle. The document will be of great interest to those working in the field and, once a few minor issues have been resolved, it will make an excellent contribution.

1. This may a consequence of the word limit imposed by Nat Commun, but I found the text a bit difficult to follow owing to the abundant use of parentheses and "e.g.". Please consider a readthrough for improved flow.

2. In the first paragraph on page 2, it is mentioned that medical overdosing and medical errors kills 100,000 people per year in the USA. Is this number correct? Reference [5] evaluated 450,385 medical errors of which 17,388 medication errors with adverse

patient care outcomes defined as requiring additional drug therapy are reported. Also, a more recent reference would be appreciated since [5] is dated from 2001.

3. Wrong usage of the term personalized medicine => Change for drug delivery system related terms.

The terms 'stratified,' 'personalized' or 'precision' medicine all refer to the grouping of patients based on risk of disease, or response to therapy, using diagnostic tests or techniques.

4. It would be appreciated having information on the computer software for the sequence engineering, the predicted structure, the DNA binding site of D0, D1, Q0 and Q1 to their specific ligand. This can be added in the supplements. How did the authors predict the variant buffers D1 and Q1 to have reduced affinity before determining it experimentally?

5. Page 6 "We first determined the toxicities of doxorubicin for HeLa cells (IC50 = 680 nM, Fig. 3A) and for HCT116 cells (IC50 = 528 nM, Fig. S5A), which were similar to the values

reported in the literature (20)." The present phrasing is a bit too general. It would be advisable to rephrase to specify that the results emerge from past work by the authors.

6. On page 7, does the difference in cellular localization between DNA in the cytoplasm and doxorubicin have an influence on the drug's mechanism of action? Can the authors comment on the mechanism of cellular trafficking?

7. How is it that DNA does not trigger an immune response since innate and adaptive recognition of non-self would lead to accelerated clearance giving rise, potentially, to spontaneous release of the drug?

8. Not all molecules can bind with DNA due to size constraints, charge distribution. So, is the proposed approach limited to small molecules intercalating with DNA or transcription factors such as hormones could be the subject of another study? What are the limits of your approach?

9. The references in the discussion are somewhat dated. It would be beneficial to provide a more up to date perspective on delivery systems for dox, quinine and their analogs.

Excellent paper that requires some minor corrections.

Responses to Referee

To help the reviewers, we have organized the referees' comments whereby the format is that the modification of the manuscript for comment from Referee X is highlighted with a specific color. **Our responses are shown in blue.**

Response to Referee #1: modified/added texts in the revised manuscript are shown in yellow for this referee.

1.1

The manuscript "Programmable, self-regulated molecular buffers for precise, sustained drug delivery" by A. Desrosiers and co-authors presents an **ingenious approach to controlling drug delivery *in vitro* and *in vivo* using DNA-based molecular buffers**. By tuning the design of the molecular buffer (its drug affinity, stability in serum, ability to bind albumin, etc.) as well as its formulation (drug/buffer ratio) the authors have found a **versatile strategy to control drug concentration ranges in variety of contexts for a multitude of potential drug types**. As a proof-of-concept, the authors have performed a thorough characterization of their system from molecular design to *in vivo* assessments and have made intriguing findings regarding the effect of molecular buffers on drug properties. **I heartily recommend this article for publication in Nature Communications.**

Answer: We thank Referee #1 for this efficient and precise summary of the manuscript.

1.2

While the use **DNA-based molecular buffers is very convincing to control the concentration of free drug in solution or plasma, it is unclear how it compares to other sustained drug delivery systems found in the literature**. If possible, I recommend the authors **provide more context in the introduction and conclusion on how their approach addresses challenges in the field of sustained drug delivery and what are the next challenges to address**. This will help the reader appreciate the impact of their findings.

Answer: This is a good point: we have added these additional sentences in the introduction and discussion that clarifies this point.

"Typically, oral or local sustained release is achieved using erodible or swelling polymer matrices that delay the diffusion of a large drug payload, compensating for the drug degradation/clearance (7). Injectable DDS, such as lipid or polymer-based nanomedicines, encapsulate drugs with unfavorable biopharmaceutical properties (low solubility, low permeability) and improve their bioavailability, biodistribution and usually prolong their blood circulation time (7). Unfortunately, these DDS do not take into account the individual pharmacokinetic specificities and result in significant interindividual variability in drug plasmatic concentrations (11, 12). Impressive progress has been made in the local administration of on-demand drug delivery systems, for

instance the intelligent wearable medical devices, but their development is still complex and costly (9, 13).”

1.3

Are there target values that indicate relevance for clinical applications?

Answer: This is a good point: we have added additional sentences in the introduction and discussion that clarifies this point.

“Such molecular buffer follows *Le Chatelier’s* principle, in which the equilibrium between the complexed and free thyroxine shifts in response to changes in thyroxine levels in blood to maintain its free concentration near the thyroxine-TBG dissociation constant $K_D \sim 100$ pM (16). Currently, sustained release systems do not follow this principle, since their release is not impacted by the elimination of the drug. Few self-regulated devices or platform have been developed but they have not yet reach the clinics (15). The “active” or target concentration required for the desired therapeutic effect is typically obtained through pharmacodynamics studies during preclinical and clinical studies. Then therapeutic dosage is selected and adjusted in order to reach optimal therapeutic outcome (17). However, the selection of an optimal therapeutic dosage for a new drug entity remains a major challenge since 16% of drugs fail the FDA review cycle because of uncertainties in their dosage selection (18). Strategies overcoming this challenge by delivering the drug directly at the “target concentration” would greatly improve the chances of success of new drug candidates.”

1.4

How helpful are molecular buffers towards achieving these targets?

Answer: This is a good point: we have added additional sentences in the introduction and discussion that clarifies this point.

“Strategies overcoming this challenge by delivering the drug directly at the “target concentration” would greatly improve the chances of success of new drug candidates. In addition, the ability to maintain the therapeutic drug concentration in patients displaying different pharmacokinetics would drastically reduce the interindividual variability and enlarge the target patient population. Furthermore, it would likely reduce the frequency of administration, improve patient compliance and treatment efficacy.”

1.5

Currently, the introduction and conclusion are both very narrowly focused on their design and achievements and a lacking in explicit statements of the relevance to a broader context such as what would be expected in a clinical setting or comparisons against state-of-the-art formulations for sustained delivery, which complicates assessments of novelty and relevance.

Answer: We hope that the additional information provided above provides a broader context in terms of clinical setting or comparisons versus state-of-the-art formulations.

1.6

Throughout the manuscript, the authors discuss how changing different aspects of the design and formulation of their molecular buffers result in modulation of the toxicity, reservoir quantity, stability, distribution and therapeutic/side effects of the drug released. **Currently, the manuscript is very effective at showcasing the experiments that were performed and the conclusions that were drawn.** However, many of these properties are tested separately in different contexts (in vitro vs in vivo, organ imaging vs in serum etc., some assays are done in solution with and without enzymes), which complicates the assessment of the key design principles that could be used in designing molecular buffer outside of the doxorubicin/quinine set explored. **I recommend the authors provide key take-aways or explicit guidelines as a paragraph in their discussion for readers who would be interested in applying these molecular buffers to different contexts since they are presenting their strategy as a general platform.** Is there an overall go-to recommendation for the K_D of the molecular buffer when the therapeutic window of a drug is known? Which modifications (e.g., phosphorothioate, G-quadruplex, cholesterol, C16, etc.) are more effective in promoting desired therapeutic outcomes over side effects in vivo? If these are not known, are there general recommendations for designing a DNA-based molecular buffer for a selected drug?

Answer: This is an excellent suggestion by Referee #1. We have added the following paragraph highlighted in yellow to the discussion:

“With this *in vivo* proof-of-concept showcasing its potential, we believe that molecular buffers could improve the delivery of drugs that display a small therapeutic window and/or for which selection of an optimal therapeutic dosage remains challenging. We also believe that molecular buffers could further contribute to optimize drug properties such as its log p (optimal between -0.4 to 5 (42)) its biodistribution as well as potentially reduce drug resistance by maintaining drug concentration in the target range. To facilitate the development of future molecular buffers, we propose below the following short guideline:

- 1) Find a known binding partner for the drug of interest. It could be found in the scientific literature or in a database (e.g. aptagen for aptamers and peptides) or alternatively, selected via various techniques (e.g. SELEX, phage-display...);
- 2) Measure the binding affinity (K_D) and stoichiometry between the molecular buffer and the drug and optimise its K_D to be near the target concentration. This can be realised via various strategies (43, 44) that include point mutations (45), creating a structure-switching molecular buffer and modifying its switch equilibrium via mutation (24), inhibitors or activators (45-47);
- 3) Validate that drug binding to its molecular buffer inactivates its therapeutic activity through cell culture viability tests;
- 4) Modify the molecular buffer structure to optimize its chemical stability, biodistribution, and log P . For example, we have changed the DNA backbone of our molecular buffer to phosphorothioate to reduce its degradation from nuclease and have also added a C16 anchor aimed to bind to albumin that maintains the buffer and doxorubicin in blood circulation. PEG modifications could also improve

the buffer half-life by reducing its clearance rate (48). These modifications should not significantly affect the drug binding affinity (K_D). Re-optimize K_D if needed;

- 5) Characterize the buffer and drug/buffer formulation pharmacokinetics *in vivo* to determine the efficiency of the molecular buffer;
- 6) Select an optimal formulation (drug/buffer ratio) and dosage to optimize treatment outcome.”

1.7

Minor edits suggested:

I would recommend individual labeling of panels in figures instead of in groupings by themes to help connect with the main text. (e.g., when referring to Fig. 5B, there should be a title associated to panel explaining what is found specifically in this figure panel). Similarly, matching data point labels in related figures would be appreciated (e.g., Fig 4b and c, why is only the 160 μ M data point highlighted? Fig 4e and f could also be matched similarly to facilitate figure reading).

Answer: Good idea, we have updated figure 2, 4 and 5 accordingly

1.8

Overall, the manuscript is excellent and well-written. It contains a sound methodology with well-executed experiments. The authors use a clearly outlined approach and report outcomes in a systematic manner, enabling them to comment on important design parameters for assessing the performance of DNA-based molecular buffers in the context of sustained drug delivery.

Answer: We thank Referee #1 for this efficient and precise summary of the manuscript.

Response to Referee #2: modified/added texts in the revised manuscript are shown in blue for this referee.

2.1

The study conducted by the authors is commendable as they are endeavouring to formulate self-regulated molecular buffers that would be able to regulate the desired concentration of the free drug both *in vitro* and *in vivo*.

Answer: We thank Referee #2 for this efficient and precise summary of the manuscript.

2.2

1) The authors claim that the molecular buffer is universal. The usage of the term universal can be slightly inaccurate as their formulation can never be parlayed into another system (another drug), as each new drug needs a unique interacting partner. If the authors still choose to cling on to the term universal, more clarifications are needed to justify the nature of universality of the buffer system.

Answer: We agree with R#2 that a specific DNA sequence design for a specific drug binding can only be useful for a specific drug. Thus a single DNA sequence is not itself a universal molecular buffer. In this paper, our objective was to use the term universal when referring to the concept of buffering a drug. We make the argument that in theory any medical drug could potentially be buffered (meaning bound to) by either a DNA aptamer (identified through SELEX) or a peptide/protein molecular buffer. We have introduced some clarification on the concept of universality regarding our molecular buffer.

Here is the modification that was highlighted in blue in the manuscript:

“The concept of a molecular buffer is universal and could be envisaged with any type of drug using any type of chemical receptor like DNA or proteins, provided that the molecular buffer has a specific affinity for the drug of interest.”

2.3

2) It would be great if the prowess of aptamer is accentuated by actually incorporating the term aptamer in the title. For example-‘Programmable, self-regulated molecular buffer-assisted aptamer for precise, sustained drug delivery.’

Answer: We agree with this reviewer that our title and abstract did not clearly reflect our usage of nucleic acids and aptamers. We have modified our abstract (see below) to clearly highlight our use of nucleic acids and aptamers as well as their programmable advantages. However, we strongly believe that the molecular buffers strategy represents a universal approach that can be adapted to any other polymers (e.g. peptides, proteins, sugars,...). We therefore believe that our title should remain broad enough so that our work remains of appeal for drug delivery researchers that prefer to employ other polymers than aptamers. In the case of doxorubicin, for example, we could also have employed a small peptide sequence that binds doxorubicin for our proof-of-

concept (Multidrug-resistance drug-binding peptides generated by using a phage display library. *European Journal of Biochemistry* 251, 155-163 (1998)).

Here is the modification that was highlighted in blue in the abstract:

“Unlike artificial nanosystems, biological systems are ideally engineered to respond to their environment. As such, natural molecular buffers ensure precise and quantitative delivery of specific molecules through self-regulated mechanisms based on *Le Chatelier’s* principle. Here, we apply this principle to design self-regulated **nucleic acid** molecular buffers for the chemotherapeutic drug doxorubicin and the antimalarial agent quinine. We show that these **aptamer-based buffers** can be programmed to maintain any specific desired concentration of free drug both *in vitro* and *in vivo* and enable the optimization of the chemical stability, partition coefficient, pharmacokinetics and biodistribution of the drug. These programmable buffers can be built from any polymer and should improve patient therapeutic outcome by enhancing drug activity and minimizing adverse effects and dosage frequency.”

2.4

3) Why didn't the authors use UREA-PAGE based purification to purify the oligos? UREA-PAGE based purification can be more selective as the synthesis of oligonucleotides using DNA synthesizer can result in multiple truncated forms along the way and this might affect the formulation of the self-regulated molecular buffers.

Answer: We have used reverse-phase purification on a silica-based column to purify our oligos for various reasons, we agree that urea-page based purification could also have been used.

*A little more about “reverse-phase purification”: Truncated sequences are capped during DNA synthesis while complete sequences bear a highly hydrophobic protecting group, DMT, on its 5' end. We use this DMT to perform reverse-phase purification on a silica-based column. The purification is automated and consist in multiple washing steps to remove the truncated DNA sequences followed by the elution. As mentioned by the reviewer, not all truncated sequences can be removed but sufficient purity (> 90%) can be obtained for our short doxo transporter containing only 24 nucleotides (see *Current Protocols in Nucleic Acid Chemistry* (2000) 10.7.1-10.7.5).*

Other factors that contributed to our choice include:

1) For bigger scale DNA synthesis (e.g. multiple 1 umol column), we would need much bigger gels (at least 5 mm thick) that are more complicated to run with small scale electrophoresis apparatus. Furthermore, DNA bands need to be cut under a UV lamp which could lead to some chemical modifications.

2) The yield of DNA extraction from a cartridge purification gel is higher than what we get on a gel, which is convenient when large amount of DNA are required (e.g. mice experiments).

2.5

4) The authors mentioned 'A first approach consists of varying the K_D of the buffer'. This is quite perplexing as to how one could change the K_D of a buffer which is constant. How the authors were able to vary the K_D of the buffer?

Answer: This is a very interesting point raised by Referee #2.

Perhaps our sentence "For example, buffer variants Q1 and D1 (see Fig. S3) are engineered to display lower affinities for quinine and doxorubicin" was a little bit misleading and we have modified it to make it more clear.

We have added the following modified sentence in the manuscript to clarify our strategy:

"We can also program our buffers to maintain a specific, desired concentration of free drug. A first approach consists of varying the K_D of the buffer. For the quinine aptamer, we introduced site specific mutations (see Materials and Methods) that reduce the affinity for quinine (aptamer Q1 and Q2). Since doxorubicin binds duplex DNA through intercalation in GC base pair, mutations could not be used to tune their K_D . To circumvent this limitation, we explored and found that specific G-quadruplex sequences (D1 and D2) display K_D that are 4 and 27-fold higher than the original D0 GC duplex DNA (35, 36)."

We also added more explanations on our strategy in the Materials and Methods section:

"Approach to vary the K_D of the buffers:

For quinine, our hypothesis was that site specific mutations could likely reduce the binding affinity for quinine (this strategy would likely not lead to an increase in binding affinity). We have explored various mutations (20) that were based on the three-way junction Q0 DNA aptamer. We hypothesized that quinine would bind in the three-way junction and thus modification of the stems (either stabilizing, destabilizing, increasing length, decreasing length) while maintaining the three-way junction secondary structure would yield similar or lower binding affinity for quinine. With this strategy, we had a success rate of around 30% (lots of sequence were either of similar K_D as previously obtained, unchanged K_D or K_D that were too high). Of note, this strategy remains cumbersome in absence of structural information on the aptamer and its binding site since the mutations will remain random.

For doxorubicin, we did not alter the D0 sequence since doxorubicin binding affinity for DNA duplex with GC bp doesn't vary. To circumvent this limitation, we hypothesized that G-quadruplex sequences would also bind doxorubicin since there a multiple G-G stacking which was indeed the case. The affinity of doxorubicin was also different with different G-quad sequence (they all have different structure and thus we can work with different K_D). We used sequences available in the literature."

We also added this section in the discussion to show other strategies to optimize the binding affinity if needed:

“Measure the binding affinity (K_D) and stoichiometry between the molecular buffer and the drug and optimise its K_D to be near the target concentration. This can be realised via various strategies (43, 44) that include point mutations (45), creating a structure-switching molecular buffer and modifying its switch equilibrium via mutation (24), inhibitors or activators (45-47).”

2.6

5) The authors also used the term ‘drug degradation’. First of all, the authors ought to define the definition of the word degradation. Does it refer to the random decomposition of the drug due to any enzymatic or any action exerted by the system? Or does it refer to the dissociation of the drug from its receptor, in this case aptamers?

Answer: We agree with the reviewer and have added some additional indications at specific places, in blue to make sure our use of the word “degradation” is clearer and more specific:

- 1) “To illustrate this advantage, we simulated a drug/buffer formulation degradation or elimination through serial $\frac{1}{2}$ dilutions every hour that mimics a drug/buffer concentration decrease by 50% every hour. (Fig. 4A, left panel).”
- 2) “Molecular buffers can also be programmed to generate personalized pharmacokinetic profiles by controlling their degradation or elimination rate. To demonstrate this strategy, we designed an experimental setup simulating doxorubicin degradation/elimination by the kidneys using a dialysis cassette in which the doxorubicin that has passed through the membrane outside the cassette is considered degraded/eliminated (Fig. 4B, left panel). We also programmed the buffer degradation rate using specific amounts of DNase I (Fig. S9).”
- 3) “For example, using chemically modified DNA backbones like phosphorothioate or a G-quadruplex sequence (Fig. 5A, left panel), we designed various doxorubicin DNA buffers with increased resistance to chemical degradation in mice serum that contains various nucleases (Fig. S11) (38).”
- 4) “Using HPLC fluorescence measurements, we also validated that doxorubicin alone was mostly chemically degraded after one hour when exposed to mice serum (Fig. 4A, middle panel, black line). We then showed that the doxorubicin half-life in mouse serum increases proportionally with the buffer stability (Fig. 5A, middle and right panel). For example, the phosphorothioate buffer variant, with an increased serum stability, $t_{1/2}$ =10 hrs vs 6 hrs of the unmodified buffer, led only to 55% of doxorubicin degradation after 36 hrs in serum compared to 85% and 90% when employing the unmodified D0 or no buffer, respectively (Fig. 5A, middle panel).”

2.7

6) The authors claim that chemical modifications do not significantly affect the binding affinity for doxorubicin (K_D varies from 1 μM to 3 μM) in mouse serum. How did the authors introduce G-quadruplex modification into the aptamers as the introduction of any G-rich sequence would definitely debilitate its binding affinity?

Answer: We thank this reviewer for giving the opportunity to clarify this strategy.

Perhaps our sentence “For example, buffer variants Q1 and D1 (see Fig. S3) are engineered to display lower affinities for quinine and doxorubicin” was a little bit misleading and we have modified it to make it more clear.

Since doxorubicin binds duplex DNA through intercalation in GC base pair, mutations cannot be used to tune the K_D . We found in the literature that G-quadruplex sequences also display strong affinity for doxorubicin (see reference below) and hypothesize that their unique quadruplex structure may also change the doxorubicin affinity. Indeed we explored the affinity of two distinct G-quadruplex, D1 and D2, and found that their affinity for doxo was reduced by 4 and 27-fold compared to the original D0 GC duplex DNA ($D0=0.13 \mu\text{M}$, $D1= 0.70 \mu\text{M}$, $D2=3.8 \mu\text{M}$). So the DNA sequences of the G-quadruplex buffer (D1, D2) are not variants of the original D0 buffer (see Table I for the other sequences).

Here is the modification we added in the manuscript to clarify this (in blue)

“We can also program our buffers to maintain a specific, desired concentration of free drug. A first approach consists of varying the K_D of the buffer. For the quinine aptamer, we introduced site specific mutations (see Materials and Methods) that reduce the affinity for quinine (aptamer Q1 and Q2). Since doxorubicin binds duplex DNA through intercalation in GC base pair, mutations could not be used to tune their K_D . To circumvent this limitation, we explored and found that specific G-quadruplex sequences (D1 and D2) display K_D that are 4 and 27-fold higher than the original D0 GC duplex DNA (35, 36).”

References:

- 1) Doxorubicin exhibits strong and selective association with VEGF Pu 22 G-quadruplex.
- 2) Nemorubicin and doxorubicin bind the G-quadruplex sequences of the human telomeres and of the c-MYC promoter element Pu22.
- 3) Affinity of the anthracycline antitumor drugs Doxorubicin and Sabarubicin for human telomeric G-quadruplex structures.

Response to Referee #3: modified/added texts in the revised manuscript are shown in **green** for this referee.

3.1

I really enjoyed reading and reviewing the article by A. Desrosiers and co-workers regarding a DNA drug delivery system displaying characteristics based on *Le Chatelier's* principle. The document will be of great interest to those working in the field and, once a few minor issues have been resolved, it will make an excellent contribution.

Excellent paper that requires some minor corrections.

Answer: We thank Referee #3 for this efficient and precise summary of the manuscript.

3.2

1. This may a consequence of the word limit imposed by Nat Commun, but I found the text **a bit difficult to follow** owing to the **abundant use of parentheses and “e.g.”**. Please consider a readthrough for improved flow.

Answer: We thank this reviewer for this suggestion and have made changes accordingly to the manuscript.

3.3

2. **In the first paragraph on page 2, it is mentioned that medical overdosing and medical errors kills 100,000 people per year in the USA. Is this number correct?** Reference [5] evaluated 450,385 medical errors of which 17,388 medication errors with adverse patient care outcomes defined as requiring additional drug therapy are reported. Also, **a more recent reference would be appreciated since [5] is dated from 2001.**

Answer: We thank Referee #2 for this appreciated suggestion. We further explored the literature and have found more actual references (2021).

*Here is the modification that was highlighted in **green** in the manuscript:*

“Furthermore, these DDS cannot prevent drug overdosing, which killed more than 70,000 people in the USA in 2019 (14).”

Reference:

C. L. Mattson, L. J. Tanz, K. Quinn, M. Kariisa, P. Patel, N. L. Davis, Trends and geographic patterns in drug and synthetic opioid overdose deaths—United States, 2013–2019. *Morbidity and Mortality Weekly Report* **70**, 202 (2021).

3.4

3. Wrong usage of the term personalized medicine => Change for drug delivery system related terms. The terms ‘stratified,’ ‘personalized’ or ‘precision’ medicine all refer to the grouping of patients based on risk of disease, or response to therapy, using diagnostic tests or techniques.

Answer: We thank this reviewer for clarifying this point and have change our sentence to avoid misleading the readers (see modifications in the manuscript in green)

“As we enter the era of **smart DDS** (15), novel systems solving these challenges will have a major impact on the success of medical treatments.”

“**As we enter the era of smart DDS**, we envisage that molecular buffers will provide a new generation of “smart” therapeutic tools that take into account the individual pharmacokinetic profiles of patients and help prevent medical overdosing and medication errors.”

3.5

4. It would be appreciated having **information on the computer software for the sequence engineering, the predicted structure, the DNA binding site of D0, D1, Q0 and Q1 to their specific ligand.** This can be added in the supplements. How did the authors predict the variant buffers D1 and Q1 to have reduced affinity before determining it experimentally

Answer: This is a very interesting point raised by Referee #3.

Perhaps our sentence “For example, buffer variants Q1 and D1 (see Fig. S3) are engineered to display lower affinities for quinine and doxorubicin” was a little bit misleading and we have modified it to make it more clear.

We have added the following modified sentence in the manuscript to clarify our strategy:

“We can also program our buffers to maintain a specific, desired concentration of free drug. A first approach consists of varying the K_D of the buffer. **For the quinine aptamer, we introduced site specific mutations (see Materials and Methods) that reduce the affinity for quinine (aptamer Q1 and Q2). Since doxorubicin binds duplex DNA through intercalation in GC base pair, mutations could not be used to tune their K_D . To circumvent this limitation, we explored and found that specific G-quadruplex sequences (D1 and D2) display K_D that are 4 and 27-fold higher than the original D0 GC duplex DNA (35, 36).**”

We also added more explanations on our strategy in the Materials and Methods section:

“Approach to vary the K_D of the buffers:

For quinine, our hypothesis was that site specific mutations could likely reduce the binding affinity for quinine (this strategy would likely not lead to an increase in binding affinity). We have explored various mutations (20) that were based on the three-way junction Q0 DNA aptamer. We hypothesized that quinine would bind in the three-way junction and thus modification of the stems (either stabilizing, destabilizing, increasing length, decreasing length) while maintaining the three-way junction secondary structure would yield similar or lower binding affinity for quinine. With this strategy, we had a success rate of around 30% (lots of sequence were either of similar K_D as previously obtained, unchanged K_D or K_D that were too high). Of note, this strategy remains cumbersome in absence of structural information on the aptamer and its binding site since the mutations will remain random.

For doxorubicin, we did not alter the D0 sequence since doxorubicin binding affinity for DNA duplex with GC bp doesn't vary. To circumvent this limitation, we hypothesized that G-quadruplex sequences would also bind doxorubicin since there a multiple G-G stacking which was indeed the case. The affinity of doxorubicin was also different with different G-quad sequence (they all have different structure and thus we can work with different K_D). We used sequences available in the literature.”

We also added this section in the discussion to show other strategies to optimize the binding affinity if needed:

“Measure the binding affinity (K_D) and stoichiometry between the molecular buffer and the drug and optimise its K_D to be near the target concentration. This can be realised via various strategies (43, 44) that include point mutations (45), creating a structure-switching molecular buffer and modifying its switch equilibrium via mutation (24), inhibitors or activators (45-47).”

3.6

5. Page 6 “We first determined the toxicities of doxorubicin for HeLa cells ($IC_{50} = 680$ nM, Fig. 3A) and for HCT116 cells ($IC_{50} = 528$ nM, Fig. S5A), which were similar to the values reported in the literature (20).” **The present phrasing is a bit too general. It would be advisable to rephrase to specify that the results emerge from past work by the authors.**

Answer: We thank Referee #2 for its diligent reviewing. Below is the modification that was highlighted in green in the manuscript:

“We first determined the toxicities of doxorubicin for HeLa cells with $IC_{50} = 680$ nM (Fig. 3A) and for HCT116 cells with $IC_{50} = 528$ nM, (Fig. S5A), which were similar to the values reported in previous work (27).”

3.7

6. On page 7, **does the difference in cellular localization between DNA in the cytoplasm and doxorubicin have an influence on the drug’s mechanism of action?** Can the authors comment on the mechanism of cellular trafficking?

Answer: This is a very interesting point raised by Referee #2.

We have demonstrated (at least for doxorubicin) that the buffer does not change the drug mechanism but only its free concentration. For example, in Figure 2, we have showed that the 10 μ M 0.5-D0 formulation, which is programmed to provide a free doxorubicin concentration of 1 μ M, displays:

- 1) A cell viability level matching the one using 1 μ M doxorubicin in absence of buffer (Fig. 2 A-B).*
- 2) A similar doxorubicin cellular concentration (Fig. 2C) and distribution in the nucleus (Fig. 2D-E) matching the one using 1 μ M doxorubicin in absence of buffer.*

This result was obtained even though a small fraction of D0-fluorescein molecular buffer was found in the cytoplasm and not in the nucleus (most of D0-fluorescein was outside the cell).

In the case where the drug diffuses very rapidly and efficiently across the organism (or the cell membranes), our results show that buffer distribution does not alter the free concentration of drug even though a variation in drug/buffer ratio at a specific location is expected to create a local variation of free drug. In contrast, if the drug diffuses slowly across the membranes, a more “diffusible” buffer molecule will improve drug diffusion across all the organism. It would be interesting to explore further this hypothesis by studying the impact of drugs that display different internalization/diffusion kinetics.

Concerning the cellular trafficking, as mentioned above, specific distribution of buffers (e.g. inside or outside the cell) should have few impact on the concentration of a rapidly diffusing drug that is in a rapid equilibrium with its buffer. We believe that in such case variation, in buffer distribution will rather lead to a variation in localization of inactive drug reservoir, which should, in principle, not impact the outcome of the treatment. For

example, we believe that our C16 buffer maintained the drug concentration in the whole organism near 1 μ M even though most of the drug reservoir was localised in blood circulation. In some other specific application (e.g. for drugs with slow diffusion/penetration), it might be preferable to target the buffer at a specific location (e.g. maximize cellular intake) by using various tagging strategies. For example, an addition of cell penetrating peptide, has been show to increase DNA uptake by cell (see reference 53).

We added a sentence in the discussion to express this idea:

"The buffer could also be employed with drugs that are poorly internalized by the cell, since various modifications could alter the cellular trafficking of the buffer and its bound drug. For example, cell-penetrating peptides increase oligonucleotides uptake by the cell (55) and other modifications can target specific cellular pathways like the furin or transferrin pathways for example (56)."

3.8

7. How is it that DNA does not trigger an immune response since innate and adaptative recognition of non-self would lead to accelerated clearance giving rise, potentially, to spontaneous release of the drug?

Answer: This is a very interesting point raised by Referee #3.

DNA as well as artificial DNA induce a wide variety of immune responses of varying intensity depending on a multitude of factors like sequence length/composition, methylation status, nuclease resistance, etc. Artificial DNA are less characterized owing to the large number of available modifications. In our case, some sequences containing phosphorothioate DNA are characterized for their interactions with the immune system. It is known that TLR-9, that are part of the innate immune system, can be activated by PS-oligos depending on their sequence. This activation is even used as adjuvant in vaccine since it improves the immunization against the antigen of the vaccine (see reference 1 and 2 below). Other publications have reported that the immune system can recognise artificial oligonucleotides based on their nuclease resistance and thus longer circulation time (see reference 3 below).

In our case, we only performed a single injection that was not sufficient to induce any immunogenic response within the pharmacokinetic time-frame. However, it is an interesting question since molecular buffers could be used in a multiple injections dosage (over days or weeks for example). In that case, it remains to be tested whether such immunogenic response would cause a spontaneous release of the drug. On the other hand, there are various way to circumvent this potential problem. For example, mRNA that are much prone to immunogenic reactions were adapted by incorporating pseudouridine which greatly reduce the immunogenicity (see reference 4 below). Other methods could also be used to shield a molecular buffer from immunogenic reaction if needed like incorporated in a silica microparticle for example.

Also in our work, we should also take into account that our molecular buffer contained a modification comprised of a C16 that promotes its binding to albumin which could lead to a reduction in the TLR-9 activation which requires cell internalisation.

References:

- 1) A Novel Function of Phosphorothioate Oligodeoxynucleotides as Chemoattractants for Primary Macrophages
- 2) Phosphodiester backbone of the CpG motif within immunostimulatory oligodeoxynucleotides augments activation of Toll-like receptor 9
- 3) Immunogenic duplex nucleic acids are nuclease resistant
- 4) Incorporation of Pseudouridine Into mRNA Yields Superior Nonimmunogenic Vector With Increased Translational Capacity and Biological Stability

3.9

8. Not all molecules can bind with DNA due to size constraints, charge distribution. So, **is the proposed approach limited to small molecules intercalating with DNA or transcription factors such as hormones could be the subject of another study?** What are the limits of your approach?

Answer: This is an interesting question raised by Referee #3.

1) DNA aptamers can bind other types of targets than small molecules and are thus not limited to small molecules.

- *In our manuscript, we only used DNA aptamers because they were readily available in the literature. While most DNA aptamers bind to small molecules, there are multiple examples of aptamers binding to protein.*
- *SOMAmers made from modified nucleic acid have also been identified to bind to over 3,000 human proteins encompassing major families such as growth factors, cytokines, enzymes, hormones, and receptors. To highlight this point, we added a relevant reference in the text – see modification below).*
- *As you mentioned, transcription factors could also be employed as a drug while being transported by their specific DNA binding sequences (Vallée-Bélisle et al. **J. Am. Chem. Soc.**133 13836-9 (2011)).*

Modification:

“As our proof of concept buffer, we have employed nucleic-acid chemistry given its high convenience of easy synthesis, modification and programmability, its biocompatibility and **its relative universality to bind any molecular target (49-51).**”

2) Other types of biomolecules could be used as molecular buffer

- *For example, peptides are biocompatible and can bind a wide variety of targets including small molecules.*
- *The problem with peptides is that they are more complicated to work with than DNA since DNA nanotechnology is much more understood with engineering software like mfold widely available (not the case for peptides).*

- *Although, for selected labs with the expertise, phage-display is a technique that could in theory generate a peptide binding any target of interest in a similar way to SELEX which could expand greatly to number of available binding target.*

3) Limits of our approach

- *First, a limit of our approach is that we must fulfill two prerequisites to have a molecular buffer*
 - 1) *Binding to its target molecule*
 - 2) *Inactivation of its biological activity (thus bound fraction has no biological activity which can be more complicated for large drugs like proteins)*
- *As with drugs, buffers can also suffer from these other limitations:*
 - 1) *Biocompatibility*
 - 2) *Solubility*
 - 3) *Available large-scale synthesis*
 - 4) *Relevant half-life of the molecular buffer*
 - 5) *K_D that is optimal for the therapeutic window*

3.10

9. The references in the discussion are somewhat dated. It would be beneficial to **provide a more up to date perspective on delivery systems for dox, quinine and their analogs**

Answer: We have updated numerous references with more up to date research work.

REVIEWERS' COMMENTS

Reviewer #1 (Remarks to the Author):

The authors have addressed all of my comments with thorough answers. The comments in 1.3 and 1.4 are particularly welcome additions that I think will strengthen the relevance and broad interest of the manuscript. I recommend that the manuscript is published with the edits from the authors as it will make an excellent contribution to the field.

Reviewer #2 (Remarks to the Author):

The authors have made sufficient amendments to the manuscript to the point that it is eligible to have a slot in the prestigious journal Nature Communications. I strongly recommend that the manuscript is sanctioned for publication in the journal of Nature Communication.

Reviewer #3 (Remarks to the Author):

I am satisfied with the revisions made to the manuscript in response to my comments. I recommend publication.